# Moiré magnetism in CrBr$_3$ multilayers emerging from differential strain

**Fengrui Yao** [1,2] ✉, **Dario Rossi** [3], **Ivo A. Gabrovski**[1], **Volodymyr Multian** [1,2,4], **Nelson Hua** [5], **Kenji Watanabe** [6], **Takashi Taniguchi** [7], **Marco Gibertini** [8,9], **Ignacio Gutiérrez-Lezama** [1,2], **Louk Rademaker** [1] ✉ & **Alberto F. Morpurgo** [1,2] ✉

Interfaces between twisted 2D materials host a wealth of physical phenomena originating from the long-scale periodicity associated with the resulting moiré structure. Besides twisting, an alternative route to create structures with comparably long—or even longer—periodicities is inducing a differential strain between adjacent layers in a van der Waals (vdW) material. Despite recent theoretical efforts analyzing its benefits, this route has not yet been implemented experimentally. Here we report evidence for the simultaneous presence of ferromagnetic and antiferromagnetic regions in CrBr$_3$—a hallmark of moiré magnetism—from the observation of an unexpected magnetoconductance in CrBr$_3$ tunnel barriers with ferromagnetic Fe$_3$GeTe$_2$ and graphene electrodes. The observed magnetoconductance evolves with temperature and magnetic field as the magnetoconductance measured in small-angle CrBr$_3$ twisted junctions, in which moiré magnetism occurs. Consistent with Raman measurements and theoretical modeling, we attribute the phenomenon to the presence of a differential strain in the CrBr$_3$ multilayer, which locally modifies the stacking and the interlayer exchange between adjacent CrBr$_3$ layers, resulting in spatially modulated spin textures. Our conclusions indicate that inducing differential strain in vdW multilayers is a viable strategy to create moiré-like superlattices, which in the future may offer in-situ continuous tunability even at low temperatures.

Twisted stacks of 2D materials result in the formation of moiré structures that exhibit fascinating emergent electronic phenomena. Well-known examples include flat-band superconductivity[1] and magnetism[2,3], Mott–Hubbard states[4], and spatially modulated non-collinear magnetic textures[5–9]. The physical properties of moiré van der Waals (vdW) structures depend sensitively on the twist angle (Fig. 1a), which is normally fixed at the assembly stage and cannot be further changed. Ensuring the uniformity of the twist angle over a large area, and developing strategies to continuously tune the moiré superlattice, represent major experimental challenges[10,11].

To gain additional control, it has been proposed to exploit moiré-like structures resulting from differential strain in the

[1]Department of Quantum Matter Physics, University of Geneva, Geneva, Switzerland. [2]Group of Applied Physics, University of Geneva, Geneva, Switzerland. [3]Department of Theoretical Physics, University of Geneva, Geneva, Switzerland. [4]Advanced Materials Nonlinear Optical Diagnostics lab, Institute of Physics, NAS of Ukraine, Kyiv, Ukraine. [5]Laboratory for X-ray Nanoscience and Technologies, Paul Scherrer Institut, Villigen PSI, Switzerland. [6]Research Center for Electronic and Optical Materials, National Institute for Materials Science, Tsukuba, Japan. [7]Research Center for Materials Nanoarchitectonics, National Institute for Materials Science, Tsukuba, Japan. [8]Dipartimento di Scienze Fisiche, Informatiche e Matematiche, University of Modena and Reggio Emilia, Modena, Italy. [9]Centro S3, CNR-Istituto Nanoscienze, Modena, Italy. ✉e-mail: fengrui.yao@unige.ch; louk.rademaker@unige.ch; alberto.morpurgo@unige.ch

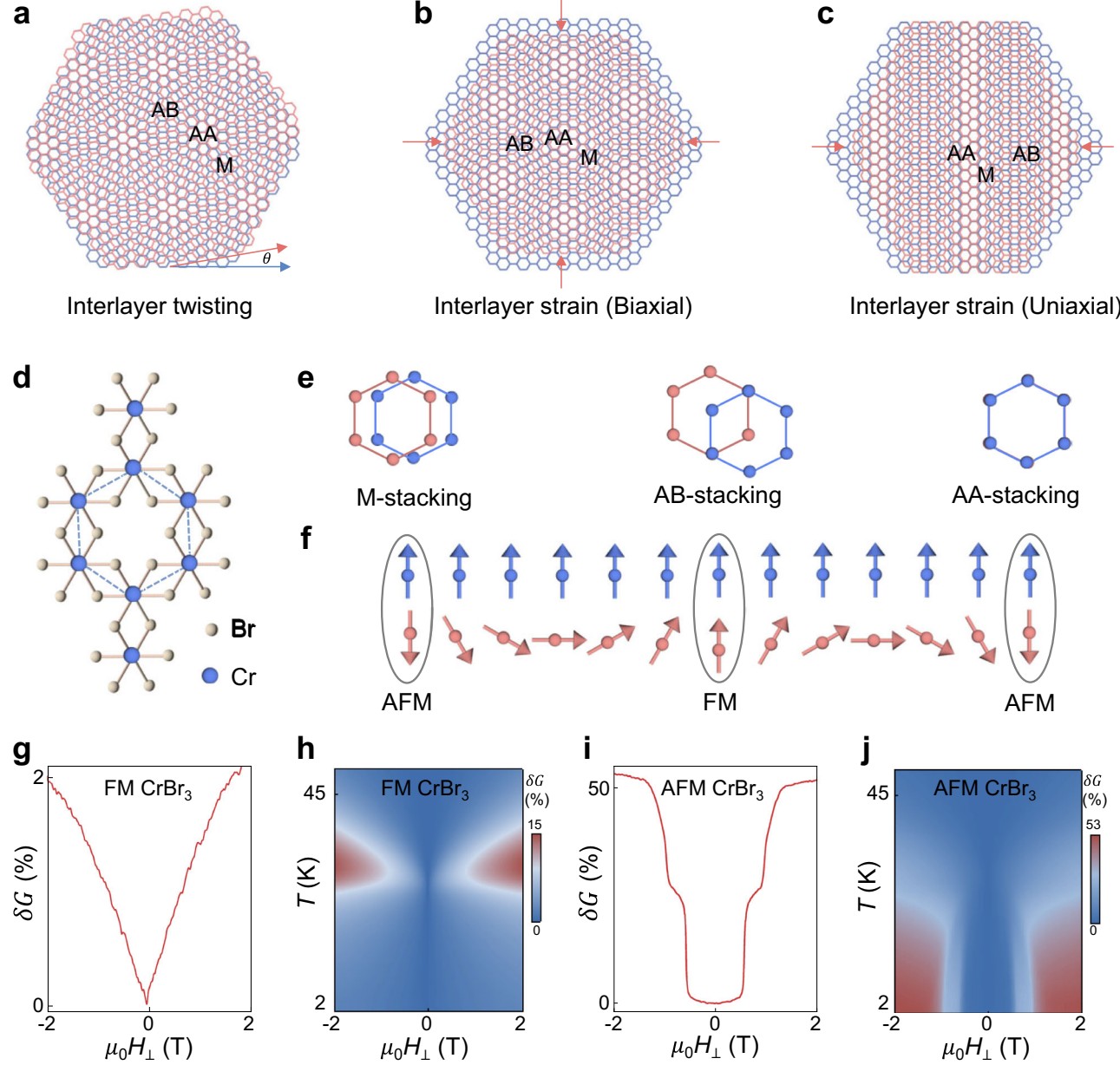

**Fig. 1 | Strain-induced moiré superlattices and stacking-dependent magnetism of CrBr₃.** **a** Moiré superlattices commonly originate from a small twist angle $\theta$ between identical vdW layers. They can also arise from interlayer biaxial (**b**) and uniaxial (**c**) differential strain, which cause adjacent layers to have slightly different lattice vectors. Red and blue honeycomb lattices represent atoms in the two layers. **d** Top view of the lattice structure of monolayer CrBr₃ and its unit cell, displaying the honeycomb lattice formed by Cr atoms (blue balls) within the edge-sharing octahedra of Br atoms (white balls). **e** Depending on how layers are stacked, three distinct (meta)stable structures of CrBr₃ are known, with different interlayer exchange coupling: the M (monoclinic) and AA stackings lead to antiferromagnetic (AFM) interlayer coupling; the rhombohedral (AB) stacking leads to interlayer ferromagnetic (FM) ordering (only the Cr atoms are depicted; red and blue balls represent Cr atoms in the two layers). **f** In differentially strained CrBr₃ layers, the spatial dependence of the interlayer exchange spontaneously results in the formation of non-collinear spin textures (i.e., moiré magnetism). **g** the tunneling magnetoconductance $\delta G(H, 2\,\mathrm{K})$ of FM barriers (data measured on a four-layer AB stacked CrBr₃ junction) is small (2%) at low temperature and exhibits characteristic "lobes" near $T_C$ as shown in **h**. In contrast, AFM barriers (data measured on a four-layer M stacked CrBr₃ junction) exhibit a large low-$T$ magnetoconductance (**i**) due to the spin-flip transitions of the inner and outer layers, which is suppressed as $T$ is increased, and which vanishes above $T_N$ (**j** see ref. 28. for the analogous data measured in an AA stacked AFM CrBr₃ barrier and for more information about the devices used to measure the data shown in this figure).

direction perpendicular to the layers[12,13]. The idea is to create a strain pattern in multilayers of vdW materials such that neighboring layers are strained differently, with the difference in lattice vectors in adjacent layers determining the resulting moiré pattern (Fig. 1b, c). This scheme mimics what happens in hetero-bilayers of semiconducting transition metal dichalcogenides, where the moiré originates from the naturally occurring difference in lattice constants[14,15]. The key advantage offered by differential strain is that its strength can in principle be varied continuously, resulting in a tunable moiré periodicity. Despite the timeliness of the subject, however, no experiments have been reported that show how the presence of a strain gradient in vdW materials creates moiré-like structures hosting phenomena analogous to those observed in twisted multilayers.

Here, we demonstrate that differential strain gives rise to moiré magnetism[5-9,16-24] in multilayers of an originally ferromagnetic system, resulting in the coexistence of ferromagnetic and antiferromagnetic regions. Our experiments rely on tunneling magnetotransport measurements through $CrBr_3$ barriers sandwiched between a $Fe_3GeTe_2$ metallic ferromagnetic electrode and a graphene contact. The magnetoconductance of such devices shows that the expected spin-vale effect –determined by the relative orientation of the magnetization in the $Fe_3GeTe_2$ electrode and the $CrBr_3$ barriers—coexists with an unexpected, reproducible background. This background is virtually identical to the tunneling magnetoconductance that we measure on small-angle twisted $CrBr_3$ barriers contacted exclusively with (non-magnetic) graphene electrodes, from which we conclude moiré magnetism is at the origin of the effect. To elucidate what causes the emergence of a moiré pattern in multilayers that are originally ferromagnetic, we perform Raman measurements showing how at low temperatures under the $Fe_3GeTe_2$ electrode, the structure of the $CrBr_3$ multilayer breaks its rhombohedral symmetry , as expected in the presence of differential strain. We complement our experiments with a theoretical analysis, which predicts that differentially strained $CrBr_3$ barriers should host a background magnetoconductance with a shape and on a magnetic field scale compatible with our experimental observations. These results demonstrate the possibility to induce moiré physics in the absence of twisting between layers, exclusively from differences in lattice parameters that originate from differential strain.

## Results

### Detecting moiré magnetism with magnetotransport

A single $CrBr_3$ layer (see Fig. 1d) is ferromagnetic with out-of-plane magnetic order, and Curie temperature near 30 K[25,26]. In the three known (meta)stable structures of the material[27,28], the coupling between adjacent $CrBr_3$ layers is either ferromagnetic –for rhombohedral (AB) stacking− or antiferromagnetic−for AA or Monoclinic (M) stacking (see Fig. 1e). As our investigations of $Fe_3GeTe_2$(FGT)/$CrBr_3$ structures rely on tunneling magnetotransport measurements, we illustrate the methodology by discussing the recently reported magnetoconductance $\delta G$ of these naturally occurring ferro and antiferromagnetic $CrBr_3$ barriers with graphene (Gr) electrodes.

Tunneling occurs in the Fowler-Nordheim regime and the magnetoconductance is due to the alignment of the spins in the $CrBr_3$ barrier, with increasing spin alignment that lowers the barrier height[28-30]. Accordingly, in ferromagnetic $CrBr_3$ barriers the magnetoconductance is small at low $T$ (Fig. 1g)−because the spins already align spontaneously in the absence of an applied field $\mu_0 H$−and peaks near the Curie temperature (Fig. 1h)−where the magnetic susceptibility tends to diverge[30]. In the antiferromagnetic phases of $CrBr_3$, instead, the low-$T$ magnetoconductance is large (see Fig. 1i, j), because the applied field flips the magnetization of individual layers and drastically improves spin alignment[30-35]. For both ferro and antiferromagnetic $CrBr_3$ barriers, the evolution of the tunneling magnetoconductance with $H$ and $T$ correlates to the magnetization of the barrier, and for ferromagnetic $CrBr_3$ barriers, $\delta G$ has been shown to be a function of $M$ (i.e, $\delta G (H, T) = \delta G (M (H, T)))$[30].

If one of the graphene electrodes is substituted with a FGT multilayer[36] (Fig. 2a, b), the behavior of the low-$T$ magnetoconductance changes qualitatively. When electrons are injected from FGT (Fig. 2d), hysteresis appears and the barrier conductance is smaller when the magnetization directions in $CrBr_3$ and FGT are antiparallel (Fig. 2f and Supplementary Fig. 1). The phenomenon is the expected spin-valve effect[37], as the $CrBr_3$ barrier spin-filters the spin-polarized electrons injected from the ferromagnetic contact[38]. Unexpectedly, however, the hysteretic contribution is superimposed onto a positive magnetoconductance background absent in devices with only graphene contacts. The background ($\delta G_{bg}$, bottom panel of Fig. 2f)

resembles the magnetoconductance of antiferromagnetic $CrBr_3$ barriers: it occurs on comparable magnetic field scales, has smaller but comparable magnitude, and an identical temperature dependence (compare with Fig. 1i, j, and discussion of Fig. 4), albeit without equally sharp jumps. When electrons are injected from the graphene electrode (Fig. 2e), hysteresis is nearly absent, but the magnetoconductance background remains unchanged (Fig. 2g). Virtually identical behavior has been seen in all four FGT/$CrBr_3$/Gr junctions that we have studied experimentally (see also Supplementary Fig. 4, which shows that at $T = 2$ K, the background $\delta G_{bg.}$ is on average one order of magnitude larger than the magnetoconductance of a ferromagnetic $CrBr_3$ barrier).

The absence of hysteresis when electrons are injected from graphene is understandable, because in the Fowler-Nordheim regime the resistance is dominated by the electron injection process. Hysteresis is therefore not expected when injecting from graphene, as graphene injects spin-unpolarized electrons. Following the same logic, finding that the background magnetoconductance is the same irrespective of the injecting electrode indicates that the phenomenon does not originate from injection at either contact, but is a manifestation of a property of the $CrBr_3$ barrier itself, which is modified by the presence of the FGT electrode.

To confirm that the background magnetoconductance only occurs in the presence of FGT electrodes we fabricated a pair of tunnel junctions on the same $CrBr_3$ multilayer, separated by only 2–3 microns. In one of the junctions, the $CrBr_3$ barrier is sandwiched between two graphene contacts, and in the other junction, one of the electrodes is an FGT crystal (see Fig. 3a). As expected, the magnetoconductance measured on the junction with two graphene contacts shows the typical behavior of ferromagnetic $CrBr_3$ barriers: very small magnetoconductance at low-$T$, Fig. 3b, and "lobes" near $T_C$, Fig. 3c (compare with Fig. 1g,h), confirming that the multilayer is indeed ferromagnetic[28,30]. The nearby junction realized with one FGT contact (Fig. 3d), instead, shows spin-valve effect when injecting electrons from FGT (evidence for ferromagnetism in $CrBr_3$), coexisting with the magnetoconductance background described above, which persists when injecting electrons from graphene (Fig. 3e). Again, the temperature evolution of the magnetoconductance background (Fig. 3f) resembles that measured in antiferromagnetic $CrBr_3$ tunnel barriers (compare with Fig. 1i, j), with all features shifting to lower field as temperature is increased and disappearing above $T_C$. Note that the background coexists with the positive magnetoconductance "lobes" above $T_C$ typical of ferromagnetism[30].

These observations establish that whenever FGT contacts are used the magnetoconductance systematically exhibits a magnetic field and temperature dependent background that is indicative of the presence of antiferromagnetism, even if a purely ferromagnetic pristine $CrBr_3$ multilayer is employed to realize the tunnel barrier. This is confirmed by a second device with the same geometry, which exhibits virtually identical behavior (Supplementary Fig. 2). We therefore conclude that bringing a $CrBr_3$ ferromagnetic multilayer into contact with a FGT electrode induces antiferromagnetism in $CrBr_3$. Ferro and antiferromagnetic regions are then simultaneously present, as expected in the presence of a moiré, and such coexistence can account for all the different aspects of the measured magnetoconductance.

### Magnetoconductance of twisted $CrBr_3$ tunnel barriers

To confirm that the coexistence of ferromagnetism and antiferromagnetism measured in FGT/$CrBr_3$/Gr tunneling junction comes from moiré magnetism, we have compared the behavior of these devices to that of small-angle (less than 3°) twisted $CrBr_3$ barriers, similar to twisted $CrI_3$ bilayers in which moiré magnetism is established[5-8]. Three twisted barriers were fabricated employing a common tear-and-stack process (see Methods for detail), to assemble two ferromagnetic $CrBr_3$ multilayers (-10 nm thick) on top of each

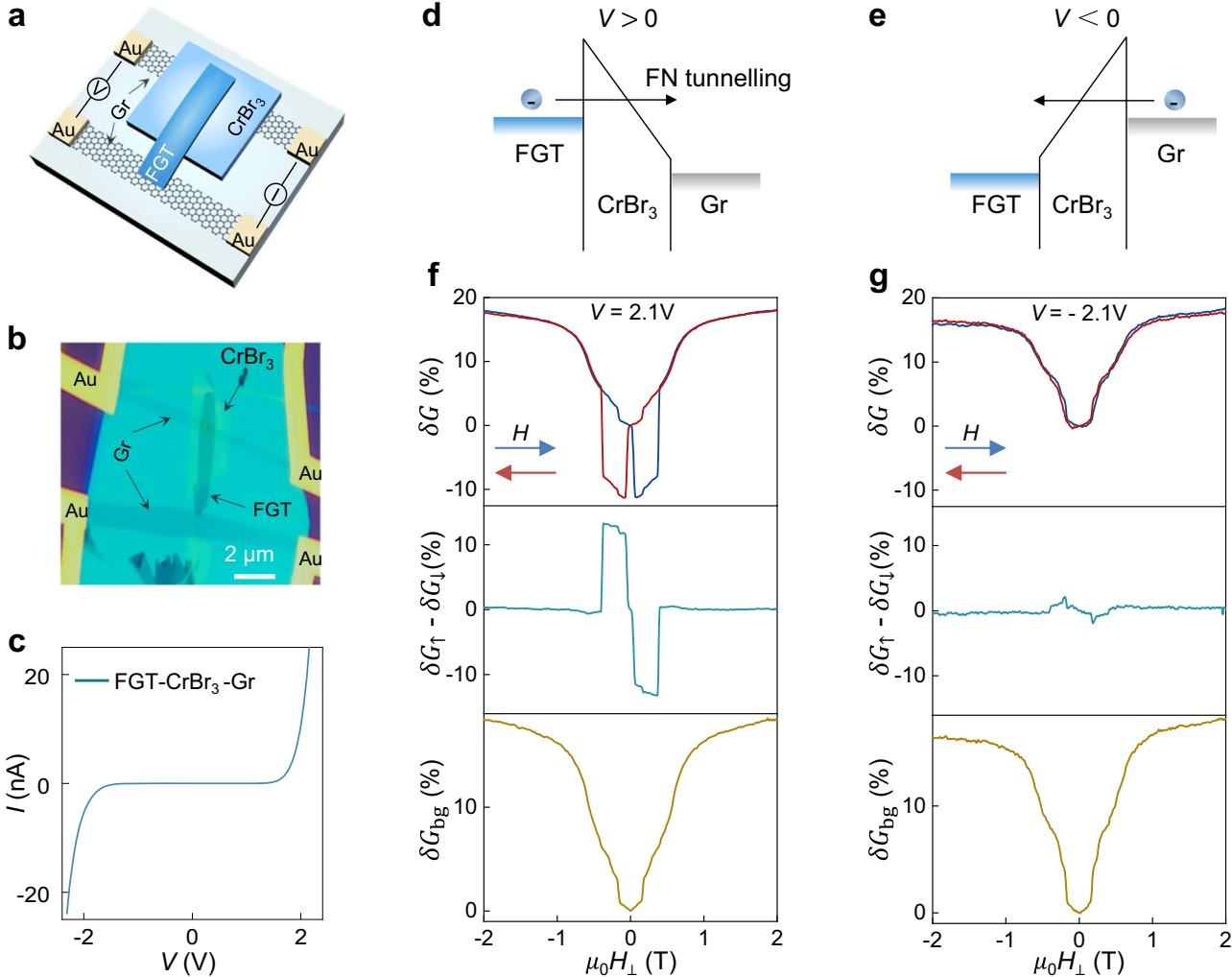

**Fig. 2 | Coexistence of ferro- and antiferromagnetism in Fe$_3$GeTe$_2$(FGT)/CrBr$_3$ barriers.** Schematic structure (**a**) optical micrographs (**b**) and an example of current-voltage (*I-V*) characteristics (**c**) of a FGT/CrBr$_3$/graphene (Gr) tunnel barrier device (data measured at *T* = 2 K). **d**, **e** for positive and negative bias, electrons injected from the FGT or the Gr electrode respectively, tunnel through the CrBr$_3$ barrier (~8.5 nm), with transport occurring in the Fowler-Nordheim (FN) regime. **f** Tunneling magnetoconductance $\delta G(H, 2\,\mathrm{K})$ for electrons injected from the FGT contact (top panel; *V* = 2.1 V; the blue and red curves are the magnetoconductance $\delta G_\uparrow$ and $\delta G_\downarrow$ measured when sweeping the magnetic field in the direction indicated by the arrows). The hysteresis is a manifestation of the spin-valve effect, resulting in a larger (smaller) conductance when the magnetizations of FGT and CrBr$_3$ are parallel (antiparallel) to each other. The spin-valve magnetoconductance ($\delta G_\uparrow$

- $\delta G_\downarrow$, middle panel) is superimposed on a sizable magnetoconductance background ($\delta G_{bg} = ((\delta G_\uparrow + \delta G_\downarrow)+(|\delta G_\uparrow - \delta G_\downarrow|))/2$, bottom panel) that resembles the magnetoconductance measured in AFM CrBr$_3$ barriers (compare to **b**). **g** Tunneling magnetoconductance measured with electrons injected from the graphene electrode (top panel; *V* = −2.1 V). The spin valve effect (middle panel) is absent but the background magnetoconductance (bottom panel) is virtually identical to that measured when injecting electrons from the FGT contact. The observation of spin-valve effect and of the magnetoconductance background in a same device provide direct evidence for the coexistence of FM of AFM regions in the CrBr$_3$ barrier. The same behavior has been observed in all tunnel barriers that we realized with FGT contacts (in all measurements, the magnetic field is applied perpendicular to the layers).

other with a nominal twist angle of 2.5°, 2°, and 1.5°, respectively. These twist angles are within the range for which moiré magnetism is expected for Chromium trihalides[5–8]. The twisted CrBr$_3$ multilayers are sandwiched between graphene contacts. In one device, the non-twisted region was also sandwiched by two graphene contacts (see Fig. 4a) to confirm that the constituent CrBr$_3$ multilayers consist of rhombohedral ferromagnetic stacking. The results of the low-temperature magnetoconductance measurements of non-twisted and twisted regions are shown in Fig. 4b, c, respectively.

In all twisted multilayer devices, a positive magnetoconductance background nearly saturating at (or just below) 1 T is observed at *T* = 2 K, whose shape is very similar to the magnetoconductance background seen in FGT/CrBr$_3$/Gr devices (compare Fig. 4c with Fig. 2f, g bottom panels and Fig. 3e). No sharp jumps but smooth

shoulders are present, in contrast with the CrBr$_3$ antiferromagnetic barriers, in which sharp jumps are in general seen at approximately 0.2 T and 0.4 T or at 0.55 and 1.1 T depending on the specific antiferromagnetic stacking[28]. The magnetoconductance background of the twisted CrBr$_3$ devices is nearly symmetric upon reversing the applied bias, analogous to the behavior of FGT/CrBr$_3$/Gr devices. Magnetoconductance data measured upon increasing the temperature for one device (plotted in Supplementary Fig. 3) show the coexistence of features originating from ferromagnetism (lobes near $T_c$) and antiferromagnetism (all features shift to lower fields as *T* increases and disappear as *T* reaches $T_C$), closely matching the evolution seen in FGT/CrBr$_3$/Gr device whose data are shown in Fig. 3f.

To better compare the magnetoconductance curves of the four different FGT/CrBr$_3$/Gr barriers with that of the twisted CrBr$_3$ barriers,

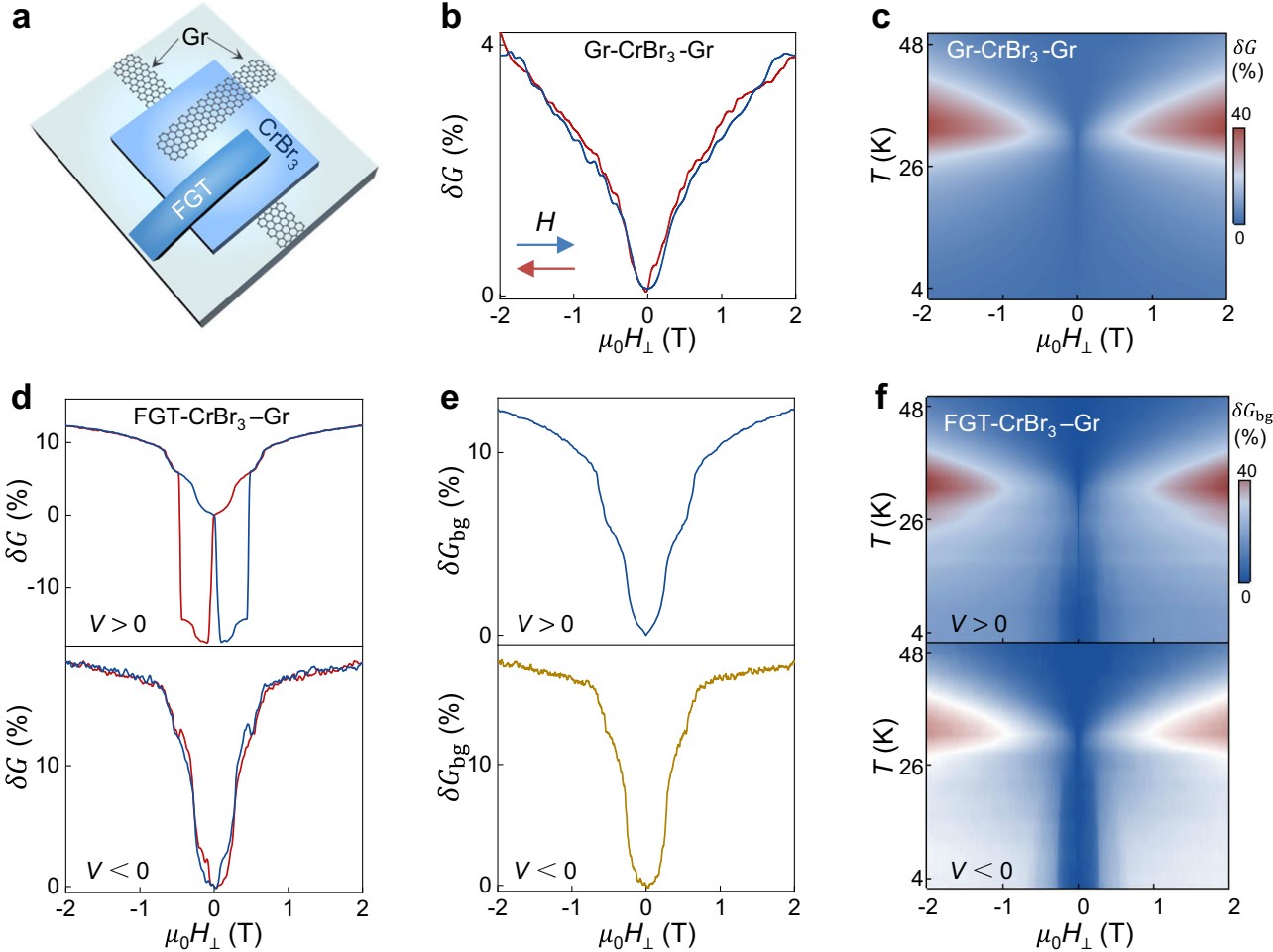

**Fig. 3 | Magnetoconductance of nearby FGT/CrBr₃ /Gr and Gr/CrBr₃/Gr junctions. a** Schematic view of a device consisting of a FGT/CrBr₃/Gr and a Gr/CrBr₃/Gr tunnel barriers realized on a same CrBr₃ multilayer (~3.4 nm), at a few micron distance from each other (h-BN encapsulating layers not shown). **b**, Tunneling magnetoconductance of the Gr/CrBr₃/Gr barrier and (**c**) color plot of its temperature dependence: the small low-temperature magnetoconductance and the "lobes" near $T_c$ confirm that the CrBr₃ multilayer is fully ferromagnetic when not in contact with a FGT crystal (compare with Fig. 1g, h). **d** Tunneling magnetoconductance of the FGT/CrBr₃/Gr junction with electrons injected from the FGT ($V > 0$, top panel) and the Gr ($V < 0$, bottom panel) electrode (data taken at $T = 2$ K). Spin-valve magnetoconductance is observed (only when injecting from FGT) and indicates the presence of ferromagnetism in the CrBr₃ multilayer. The magnetoconductance

background−present irrespective of injecting electrode (see top and bottom plots of the (**e**))−indicates the simultaneous presence of antiferromagnetism. The comparison of the magnetoconductance measured on the two nearby junctions therefore confirm that the coexistence of ferro and antiferromagnetism occurs exclusively in the CrBr₃ multilayer under the FGT crystal. **f** The color plot of the temperature-dependent magnetoconductance background extracted from the FGT/CrBr₃/Gr junction (electrons injected from the FGT (top panel) and from Gr (bottom panel)) further confirms the coexistence of ferromagnetism: the "lobes" near $T_c$ illustrate the presence of ferromagnetism, and the background shrinking in magnetic field as $T$ approaches $T_c$ (and disappearing for $T > T_c$) originates from the presence of antiferromagnetism.

we normalized the data to the value of the magnetoconductance measured at $\mu_0 H = 1$ T and plotted all curves together (see Fig. 4d). The differences in magnetoconductance between twisted CrBr₃ and FGT/CrBr₃/Gr devices is within the spread of the curves due to differences in twist angle or in strain orientation (i.e., the relative orientation of the crystalline structures of the FGT and CrBr₃ multilayers). Finding that the magnetoconductance of FGT/CrBr₃/Gr devices exhibits trends identical to those of devices based on twisted CrBr₃, whose magnetoconductance is due to moiré magnetism, confirms that –despite the absence of any twist−moiré magnetism is present FGT/CrBr₃/Gr devices.

**Probing the structure of CrBr₃ under the FGT contact**
To understand why FGT/CrBr₃/Gr devices exhibit moiré magnetism in the absence of any twist between the CrBr₃ layers, we performed Raman spectroscopy measurements to probe the structure of the

CrBr₃ multilayer under a FGT contact. In CrBr₃ −as in all other common Chromium trihalides−Raman spectroscopy can discriminate between the naturally occurring AB-stacking of the constituent layers (i.e., rhombohedral, with three-fold rotation symmetry) leading to ferromagnetism, from the monoclinic staking of the most common antiferromagnetic state, which breaks three-fold rotation symmetry. Indeed, Raman measurements are expected to exhibit a dependence on the polarization of the incident and detected light in the monoclinic structure[22,39–41], absent in the rhombohedral one. The measurements focused on the modes within the 135 –165 cm⁻¹ range, known to be sensitive to the stacking configuration, and were performed in the crossed (XY configuration, Fig. 5a) and parallel (XX configuration, Fig. 5b) polarization channels of the incident and detected light (see "Methods" Section for details).

At 20 K, Raman spectra of CrBr₃ away from the FGT contact (illustrated in Fig. 5a, b by three representative green and red curves

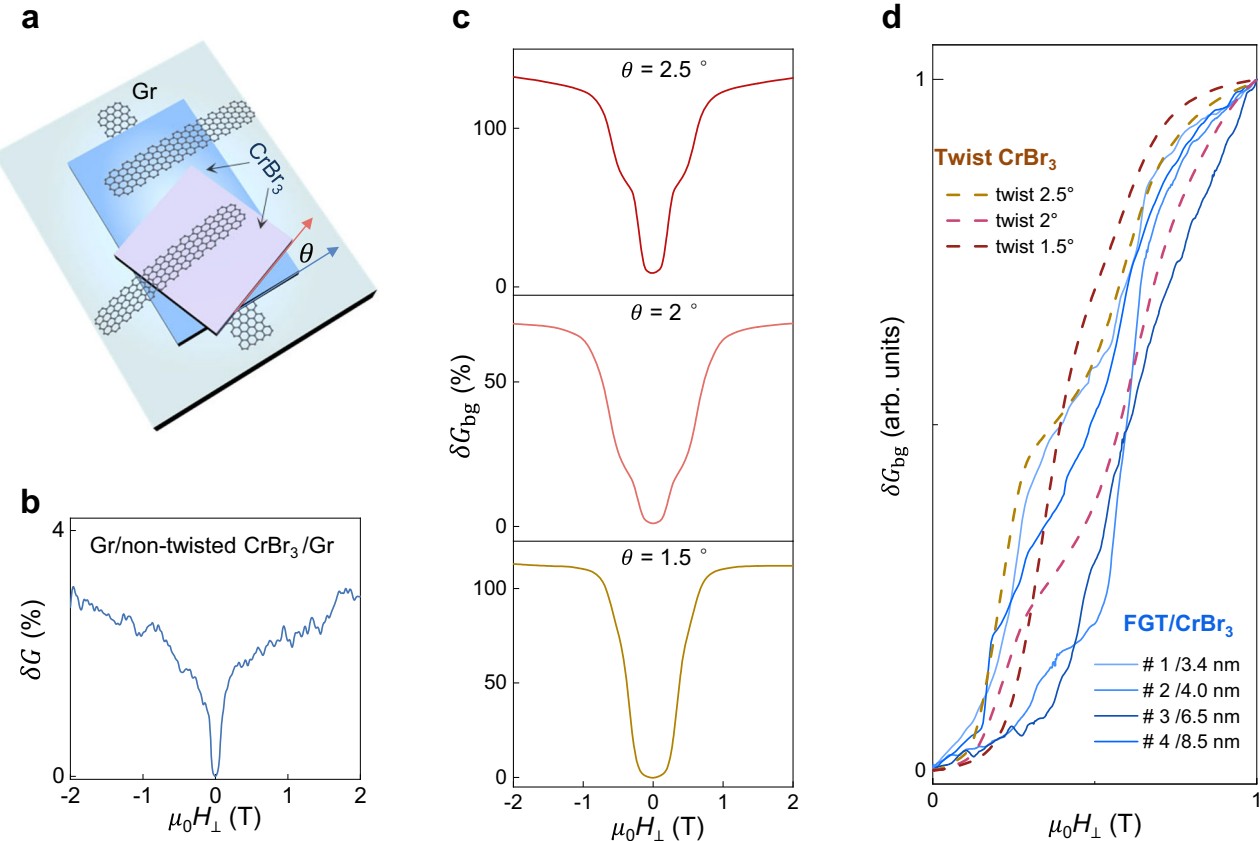

**Fig. 4 | Magnetoconductance of small-angle twisted multilayer CrBr₃ devices.**
**a** Schematic of the device configuration, showing a Gr/twisted CrBr₃/Gr junction with a Gr/untwisted CrBr₃/Gr tunnel barrier fabricated on the same CrBr₃ multilayer (approximately 10 nm thick) at a few microns' distance from each other (h-BN encapsulating layers not shown). Three devices were realized using a tear-and-stack technique to control the relative angle (θ) of the two CrBr₃ multilayers.
**b** Magnetoconductance of the Gr/non-twisted CrBr₃/Gr junction measured at $T = 2$ K, showing the behavior typical of ferromagnetic CrBr₃ barriers.
**c**, Magnetoconductance background of the three Gr/twisted CrBr₃/Gr junctions

with twist angle 2.5° (top panel), 2° (middle panel), and 1.5° (bottom panel). The magnetoconductance background is 2-to-4 times larger than that of FGT/CrBr₃/Gr barriers but otherwise shows nearly identical behavior. **d** Plot of the magneto-conductance background $\delta G_{bg}$ of all the measured FGT/CrBr₃/Gr devices (continuous lines) compared to the magnetoconductance of the three twisted CrBr₃ devices (dashed lines). All curves are normalized to 1 at $\mu_0 H_\perp = 1$ T for ease of comparison. The comparison illustrates the similarity between the curves measured in twisted CrBr₃ devices and FGT/CrBr₃/Gr devices.

measured on each side of the FGT electrode, see Supplementary Fig. 5 for detailed positions) reveal two peaks at ~142 cm⁻¹ and 152 cm⁻¹, corresponding to twofold degenerate $E_g$ modes[42]. The peak positions and intensities are the same in the two polarization channels, as expected for the rhombohedral (ferromagnetic) stacking of CrBr₃. In contrast, under FGT, we observe a broadening of the peaks, whose shape suggests the presence of overlapping peaks from multiple stackings (the Raman signal is weaker–because the CrBr₃ multilayer is located under the metallic FGT crystal–which makes it difficult to fully resolve the splitting). More importantly, the peak positions of the two Raman modes exhibit an unambiguous dependence on the polarization channel employed for the measurements, i.e., the peak positions differ for measurements done in the XX and XY polarization (Fig. 5c, d). Both the splitting of the peaks and the sensitivity to the polarization channel are distinct signatures of monoclinic stacking in CrBr₃. Their observation confirms that under the FGT electrode the structure of the CrBr₃ multilayer is modified from the common rhombohedral (ferromagnetic) stacking of CrBr₃ multilayers, as expected in the presence of a moiré.

We attribute the presence of the moiré identified by the Raman measurements–and responsible for the coexistence of ferro and antiferromagnetism in CrBr₃ in contact with FGT–to differential strain in the CrBr₃ multilayer. To explain its presence, we propose a scenario in which differential strain originates from the different thermal

expansions of FGT and CrBr₃[42,43]. In simple terms, the FGT/CrBr₃/Gr structures are assembled at room temperature, where coupling between FGT and CrBr₃ is established. Upon cooling, the difference in thermal expansion of the two materials imposes a strain in the upper layers of CrBr₃ that propagates in the layers further away. As a result, differential strain appears in CrBr₃, resulting in the appearance of moiré magnetism with coexisting of FM and AFM regions. Consistently with this scenario, we find that at room temperature no Raman shift due to a splitting nor any polarization dependence is observed (see Supplementary Fig. 5), as the shift and the polarization dependence evolve gradually and continuously upon cooling, becoming sizable only well below 100 K (see Supplementary Fig. 6, the data show no indication of sharp changes associated to a structural or magnetic phase transition).

## Theoretical analysis of strain-induced moiré magnetism
Having concluded experimentally that the magnetoconductance measured in FGT/CrBr₃/Gr devices originates from differential strain in CrBr₃ induced by the contact with FGT, we analyze theoretically the magnetic states that are expected to emerge in the presence of a strain-induced moiré. The moiré pattern that appears when two neighboring layers experience differential strain is illustrated in Fig. 1b, c. The moiré causes the stacking to depend on position, which

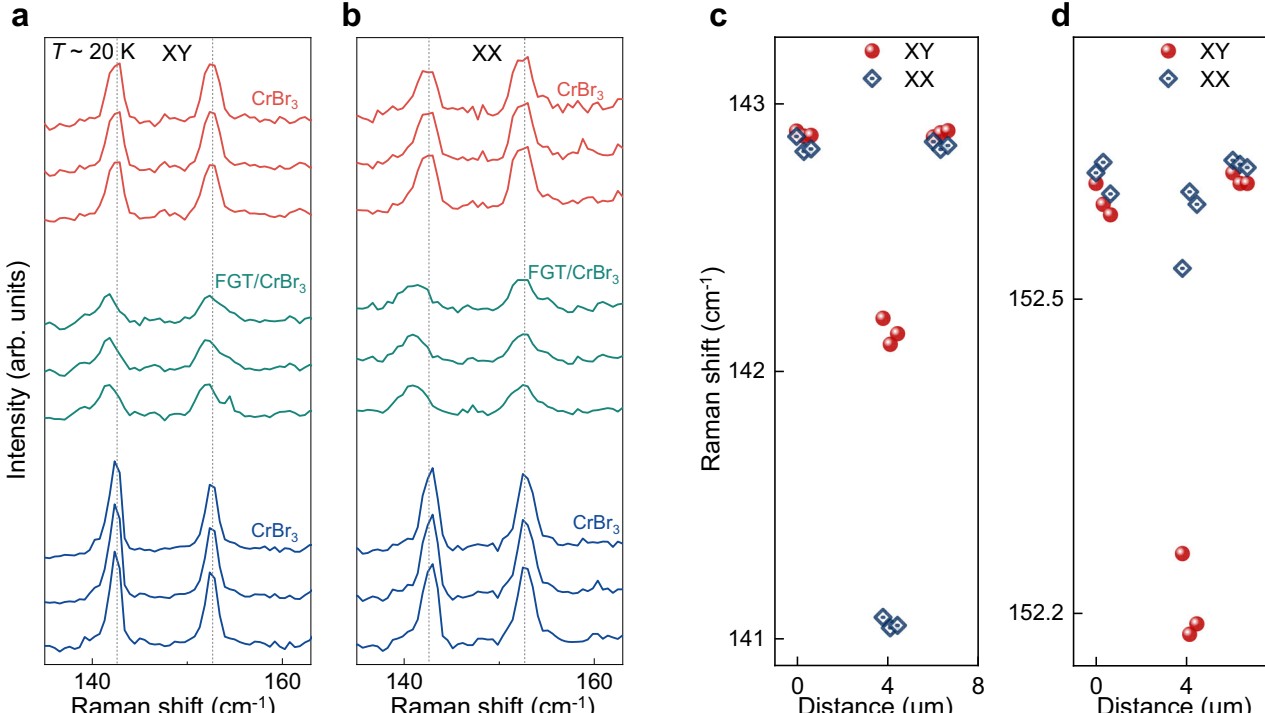

**Fig. 5 | Comparison of CrBr₃ Raman spectra next to and under a FGT crystal.** To gain information about the influence of an FGT contact on the structure of the underlying CrBr₃ multilayer (magnetoconductance in Fig. 3), the Raman spectra of the CrBr₃ tunnel barrier next to and under the FGT electrode were compared. We focused on the two $E_g$ modes near 140 cm⁻¹ and 150 cm⁻¹, known to be highly sensitive to the symmetry of the CrBr₃ structure. Raman spectra of CrBr₃ measured at different points of the CrBr₃ tunnel barrier, under crossed (XY, **a**) and parallel (XX, **b**) polarization configuration of the incident and detected light (data measured around 20 K). The red and blue curves are the Raman spectra measured at positions next to the FGT electrode (on opposite sides); the green curves are measured at different positions under the FGT electrode (see Supplementary Fig. 3 for position details). Under the FGT electrode, the peaks are broader and exhibit a shoulder (suggesting that they in fact consist of distinct peaks, i.e., that they are split). More importantly, the vertical dotted lines mark the peak position in CrBr₃ away from the FGT contact. It is apparent that under FGT the peak position is shifted. **c**, **d** Extracted peak wavelength of the two individual modes, as a function of position on the CrBr₃ multilayer where the measurements are done (on the *x*-axis, Distance = 0 μm corresponds to a position to the left of the FGT electrode; the FGT crystal is located at distances between approximately 3 μm and 5 μm). A polarization-dependent shift of the peaks measured under the FGT electrodes is apparent.

in CrBr₃ inevitably results in the simultaneous presence of interlayer ferromagnetic and antiferromagnetic domains. In practice, in our devices, the layers in contact with the FGT crystal are under strain due to the coupling between the two materials, with the strain that relaxes in the layers further away from FGT. Due to the relatively weak vdW interlayer bonding in the CrBr₃ multilayer, we expect that the combined elastic and stacking energy is lowered when all the differential strain is localized at a single bilayer moiré interface, whose exact location depends on microscopic details (see Supplementary Information Sec. 2.2 for details).

We model the presence of AFM and FM domains at this moiré interface to understand whether their coexistence can account for the experimentally observed magnetoconductance. To this end, we calculate the dependence of the magnetization $M$ on the applied magnetic field $\mu_0 H$ in the presence of many different differential strain configurations, and search for similarities between the magnetoconductance background ($\delta G_{bg}$) and the $M(H)$ curve (as mentioned earlier, in CrBr₃ barriers there is a close correspondence between tunneling magnetoconductance and magnetization[30]). For the calculations, we use a continuum field theory based on ref. 17. The magnetization of a single layer is described by a spin stiffness and a single-ion anisotropy, while the interlayer coupling is modulated throughout the moiré unit cell. Taking into account experimentally relevant parameters, we find that a CrBr₃ moiré bilayer in the absence of a magnetic field has a magnetic texture with locally *c*-axis aligned ferromagnetic or antiferromagnetic order, separated by coplanar domain walls (Fig. 6a). Upon the application of a magnetic field these domain

walls move, and the antiferromagnetic domains shrink, leading to smooth changes in magnetization that indeed mimic the observed smooth change in magnetoconductance (Fig. 6).

At a first critical field – whose value depends on the strain pattern and effective spin stiffness– the antiferromagnetic domain around the AA region disappears through a spin-flip transition (diagram I → II; Fig. 6a, b), causing a kink in the magnetization curve (Fig. 6c, pink shaded region). This is because d$M$/d$H$ is determined by the shift of domain walls, and the removal of AFM domains changes this slope. A second kink at higher fields (Fig. 6c, blue shaded region) is associated with a spin-flop (diagram II → III; Fig. 6 a, b) at a field comparable to– but somewhat larger than–the spin-flip field seen in M-stacked bilayers. This is consistent with the ab initio prediction that the strongest antiferromagnetic interlayer coupling does not occur for the M-stacked structures, but for a different monoclinic stacking (M′) that does not correspond to a (meta)stable M stacking of CrBr₃.

The features that we find are robust, in the sense that the two kinks appear regardless of the details such as the precise form of strain, stiffness, and spin anisotropy. The exact values of the spin flip and flop fields depend on details such as the thickness of the CrBr₃ multilayer and the induced differential strain, as explained in Supplementary Information Sec. 2 and Sec. 4. As shown in Fig. 6c, a qualitative agreement between the experimentally measured magnetoconductance and the square of the magnetization is found for a realistic set of model parameters (we compare to the square of the magnetization, because for ferromagnetic barriers the relation between magnetoconductance and magnetization is approximately quadratic[30]).

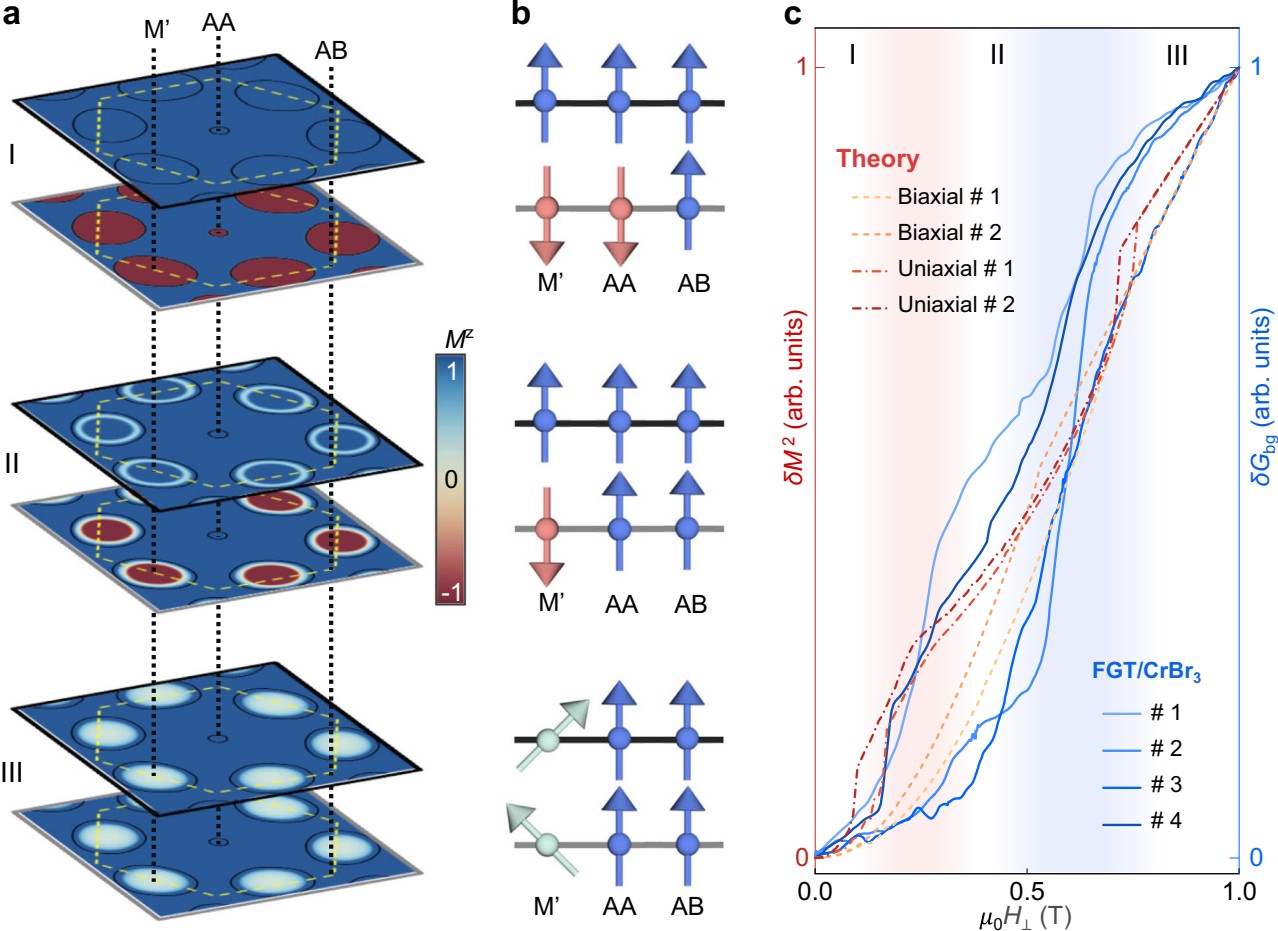

**Fig. 6 | Theoretical magnetic textures of CrBr₃ multilayers.** We analyze the measured magnetoconductance background in terms of the theoretically predicted evolution of magnetic textures in a CrBr₃ moiré interface under an applied out-of-plane magnetic field $H$. The magnetoconductance is expected to follow closely the magnetic field dependence of the square of the magnetization[30], $\delta G \sim (\delta M)^2$. **a** Visualization of magnetic textures at selected applied magnetic fields, labeled by I, II, and III. The textures are represented by the $z$-component of the local magnetization ($M^z$); yellow dashed line represents the moiré unit cell; black circles represent the boundary between ferromagnetic and antiferromagnetic interlayer Heisenberg exchange. **b** Visualization of the spin orientation in the two layers at three points in the moiré unit cell AA, AB, and M' (another monoclinic stacking at the midpoint between two neighboring AA regions). **c**, Plot of the

magnetoconductance background $\delta G_{bg}$ and of $\delta M^2$, as a function of $H$ (quantities are normalized to 1 at 1 T, to enable their comparison). Both $\delta M$ and $\delta G_{bg}$ increase smoothly at first, up to the critical field for the spin-flip transition at the AA-stacked region at $\mu_0 H_\perp \sim 0.2$ T (I → II; pink shaded region). A second smooth increase then occurs with the antiferromagnetic domains near the M'-stacked region that are further reduced in size, up to a second critical field $\mu_0 H_\perp \sim 0.5$–0.7 T associated with a spin flop-transition (II → III; blues shaded region). The two theoretical curves for biaxial strain have 1% strain, spin stiffness is 1.4 meV, and anisotropy is 0.01 meV and 0.02 meV, respectively. The two curves with uniaxial strain have 1% and 3% strain, respectively, the spin stiffness is 10 meV, and anisotropy is 0.01 meV. The overall evolution is the same irrespective of these details and reproduces qualitatively the evolution of the background magnetoconductance.

## Discussion

The background magnetoconductance of FGT/CrBr₃/Gr barriers– which exhibits the same evolution (with magnetic field, temperature and bias polarity) as the magnetoconductance of small-angle twisted CrBr₃ barrier– and the direct signature in Raman spectroscopy of the presence of monoclinic stacking in CrBr₃ under FGT provide conclusive evidence for the presence of moiré magnetism in CrBr₃ under FGT. The Raman data–which show a different peak position and polarization dependence in the CrBr₃ multilayer under and next to FGT only at low temperatures–indicate that CrBr₃ under FGT experiences strain. Taken together, these two experimental findings indicate that strain causes a moiré in CrBr₃, as expected in the presence of differential strain.

At this stage, not much can be quantitatively said about the specific properties of the strain-induced moiré at interfaces, and why the differential strain is larger for some interfaces as compared to others (for instance why it is larger at the FGT/CrBr₃ interface as compared to

the Gr/CrBr₃ interface). Since strain is likely small (1% is normally considered to be a sizable strain) and the differential strain can only be a smaller fraction of the strain of the individual layers, we expect the moiré wavelength to be much longer than in twisted structures. The data, however, does not give direct indications as to the periodicity of the strain-induced moiré. Similarly–even if Raman maps are rather homogeneous over the CrBr₃ under the FGT crystal–inhomogeneity in the moiré can be present (since the spatial resolution of Raman is diffraction-limited). Even though technically challenging–because the strained part of the structure is buried–finding ways to characterize the structural properties of the moiré present in FGT/CrBr₃/Gr devices is highly desirable. As we mentioned earlier, however, we can nevertheless establish from the temperature dependence of the Raman shift (see Supplementary Fig. 6) that strain emerges progressively as the temperature is lowered, with the Raman shift in CrBr₃ that increases gradually upon cooling, and becomes sizable only well below 100 K. That is why a scenario in which strain in CrBr₃ originates from the

difference in thermal expansion coefficients of FGT and $CrBr_3$ appears realistic.

Conceptually, the findings reported here are relevant as they show experimentally that moiré physics can indeed be accessed by inducing differential strain in multilayers of vdW materials, without the need to realize small-angle twisted structures. Creating moiré structures in this manner can be advantageous, because strain can be controlled in a variety of ways. Examples are the use of piezo actuators[44–46]—which would eventually allow strain and moiré to be tuned in-situ—of suspended structures of vdW materials[47], or possibly even of large-areas multilayers grown on suitably chosen substrates, in which the lattice mismatch determines the induced strain[48]. Inducing, controlling, and probing strain in vdW materials is a very active field of research[49,50], which will help the identification of the best-suited experimental routes to create controlled differential strain in multilayers of interest.

## Methods
### Device fabrication and measurement
The h-BN/Gr/Fe$_3$GeTe$_2$(FGT)/CrBr$_3$/Gr/h-BN and h-BN/Gr/twisted CrBr$_3$/Gr/h-BN heterostructures were assembled by means of a dry pick-up and transfer technique, employing PDMS-PC stamps within the controlled inert environment of a $N_2$-filled glove box ($H_2O < 0.1$ ppm and $O_2 < 0.1$ ppm). The FGT and $CrBr_3$ multilayers used in the experiments were obtained via micromechanical exfoliation (done inside the glove box) of bulk crystals purchased from HQ graphene. In the assembly process, a PDMS-PC stamp was used to pick up the top h-BN at 90 °C, followed by the top graphene, FGT, $CrBr_3$ and bottom graphene, each at 70 °C, and the bottom h-BN at 90 °C. The PC with the whole stack was finally released onto a $SiO_2$/Si substrate at 160 °C. After transfer, the substrate was immersed in chloroform to dissolve the PC, leaving the heterostructure on the substrate. As the FGT crystals used as electrodes are typically 10 nm thick (or somewhat thicker), air can flow between the hBN encapsulating layer and the FGT crystal edge if the edge is exposed to ambient. If so, air can reach the $CrBr_3$ multilayer causing its degradation. To eliminate these problems, we avoided etching the hBN encapsulating layer to contact directly the FGT electrode. Instead, we used separate graphite stripes connected to the FGT crystal (as detailed in Supplementary Fig. 2) and formed electrical contact to these stripes by edge contacts located far away from the FGT crystal (edge contacts were realized using electron beam lithography, reactive-ion etching, electron-beam evaporation of 10 nm Cr followed by 50 nm Au, and lift-off). For the twisted multilayer $CrBr_3$ samples, we employed the so-called 'tear-and-stack' technique. A portion of the $CrBr_3$ multilayer was picked up and stacked onto a larger, remaining layer on the substrate at a targeted twist angle ($\theta$). Designing the remaining layer larger than the picked-up portion allowed attaching graphene contacts to both the twisted and untwisted regions of the structure, which were encapsulated with h-BN on both sides.

Systematic transport measurements were conducted in an Oxford Instruments cryostat equipped with a superconducting magnet and a heliox insert. Homemade low-noise voltage bias and current measurement modules coupled with digital multi-meters were employed for the data acquisition.

### Raman Measurements
Raman spectroscopy was conducted using a Horiba system (Labram HR evolution) equipped with a helium flow cryostat (Konti Micro from CryoVac GMBH). A linearly polarized laser (532 nm, spot size ~1 μm) was focused on the sample within the cryostat through a 50X Olympus objective. The scattered light was captured by the same objective, passed through an analyzer, and directed to a Czerni−Turner spectrometer equipped with a 1800 grooves mm$^{-1}$ grating. Detection was carried out using a liquid nitrogen-cooled CCD array. By varying the half-wave plate while keeping the analyzer on the detecting light path

fixed, measurements under either parallel (XX) or crossed (XY) polarization were performed. All measured Raman spectra were fitted with a set of Voigt functions[51] (Gaussian−Lorentzian convolution) to accurately resolve the peak positions. Similarly to previous studies[39–41], the Raman tensors of the non-degenerate $A_g$ and $B_g$ modes (in stackings with broken three-fold rotation symmetry) and doubly degenerate $E_{g1}$ and $E_{g2}$ modes (in the AB and AA stacking, with three-fold rotation symmetry) of $CrBr_3$ multilayer can be written as:

$$A_g = \begin{pmatrix} a & 0 & d \\ 0 & c & 0 \\ d & 0 & b \end{pmatrix}, B_g = \begin{pmatrix} 0 & e & 0 \\ e & 0 & f \\ 0 & f & 0 \end{pmatrix}, E_{g1} = \begin{pmatrix} m & n & p \\ n & -m & q \\ p & q & 0 \end{pmatrix},$$

$$E_{g2} = \begin{pmatrix} n & -m & -q \\ -m & -n & p \\ -q & p & 0 \end{pmatrix},$$

Accordingly, for AB and AA stacked $CrBr_3$ multilayers, the Raman intensity for the $E_{g1}$ and $E_{g2}$ modes as a function of $\theta$ can be derived as: $I_{(E_{g1})} \propto |m\sin(\theta) - n\cos(\theta)|^2$ and $I_{(E_{g2})} \propto |m\cos(2\theta) + n\sin(2\theta)|^2$, where $\theta$ is the polarized direction of excitation light with respect to the analyzer. Thus, the dependence on the polarization angle cancels out when the two modes ($E_{g1}$ and $E_{g2}$) are degenerate, resulting in one single $E_g$ peak (the total intensity of the degenerate modes is the same under either XX configuration or XY configuration; observed in Fig. 4, $CrBr_3$ away from the FGT contact). However, for stackings with broken three-fold rotation symmetry, the degenerate $E_g$ modes split into the non-degenerate $A_g$ and $B_g$ modes. Thus, the position of $B_g$ mode is distinct from the $E_g$ mode and its Raman intensity as a function of $\theta$ can be expressed as: $I_{(Bg)} \propto e^2\cos^2(\theta)$, different intensities under the XX configuration and XY configuration are observed (observed in Fig. 4, $CrBr_3$ under FGT flake).

### Theoretical calculations
We compute the theoretical magnetization curves using the continuous spin model[17], with the inclusion of an out-of-plane magnetic field. The local magnetization is modeled as planar spins in the $x-z$ plane with intra-layer spin stiffness, out-of-plane single-ion anisotropy, and inter-layer Heisenberg spin exchange. For the interlayer coupling, we consider data from first-principle calculations from ref. 27. and rescale it to match the experimentally measured spin-flip critical magnetic fields for the AA and M antiferromagnetic configurations in $CrBr_3$[28]. We consider different types of moiré lattices, derived from strain and relative rotation of the layers and extract the magnetization curves from the minimization solutions as a function of the magnetic field. Further details are provided in the Supplementary Information.

## Data availability
The data generated in this study have been deposited in the Yareta repository of the University of Geneva. Source data file is provided at https://doi.org/10.26037/yareta:ftcobqbmk5bh5glrdpukq3xmuu.

## Code availability
The code adopted for calculations in this study have been deposited in the Yareta repository of the University of Geneva. Codes are provided at https://doi.org/10.26037/yareta:ftcobqbmk5bh5glrdpukq3xmuu.

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

## Acknowledgements

The authors gratefully acknowledge Alexandre Ferreira for technical support and useful discussions with Francisco Guinea, Zhe Wang, Zhu-Xi Luo, Jérémie Teyssier and Menghan Liao. A.F.M. gratefully acknowledges the Swiss National Science Foundation (Division II, project #200020_178891) and the EU Graphene Flagship project for support. M.G. acknowledges support from the Italian Ministry for University and Research through the PNRR project ECS_00000033_ECOSISTER and the PRIN2022 project E53D23001700006. L.R. and I.A.G. acknowledge the Swiss National Science Foundation (Starting grant TMSGI2_211296). K.W. and T.T. acknowledge support from the JSPS KAKENHI (Grant Numbers 21H05233 and 23H02052) the CREST (JPMJCR24A5), and World Premier International Research Center Initiative (WPI), MEXT, Japan.

## Author contributions

F.Y. and A.F.M. conceived the project. F.Y. fabricated the devices and performed the transport measurements, assisted by I.G.L; D.R., I.A.G., and L.R. performed the theoretical calculations; V.M. and F.Y. performed optical measurements; T.T. and K.W. grew and provided the h-BN

crystals. A.F.M. and I.G.L. supervised the research. F.Y., D.R., I.A.G., V.M., N.H., M.G., I.G.L., L.R., and A.F.M. analyzed the data and wrote the manuscript with input from all authors. All authors discussed the results.

## Competing interests

The authors declare no competing interests.
