## [Transparent Peer Review file · Nature Communications]

Moiré magnetism in CrBr₃ multilayers emerging from differential strain

Corresponding Author: Professor Alberto Morpurgo

Version 0:

Reviewer comments:

Reviewer #1

(Remarks to the Author)

In this work, the authors fabricate 2D tunnel junctions using a FM electrode FGT and magnetic tunnel barrier CrBr₃. Surprisingly, they find evidence for a coexistence of FM and AFM behavior in tunnel junctions using FGT, but not in tunnel junctions using only graphene electrodes (as a control) where CrBr₃ shows the expected FM behavior. This implies that the presence of FGT is somehow influencing the magnetism in CrBr₃, caused by structural effects and/or magnetic interactions.

They further use Raman spectroscopy of a CrBr₃ flake partially covered by FGT to reveal a structural difference between the bare and FGT-covered regions of CrBr₃ emerging only at low temperature, but not at room temperature. They attribute this change to an induced strain from the difference in thermal expansion coefficients between FGT and CrBr₃, which creates a local moiré superlattice with different symmetries (i.e. locally breaking the C₃ symmetry normally present in AB-stacked rhombohedral CrBr₃). Other stackings in CrBr₃ such as AA and M configurations have AFM coupling, so a strain-induced moiré structure encompassing these varied stackings would lead to the coexistence of AFM and FM order in the tunnel junction.

I agree that the data show a coexistence of FM and AFM order in the FGT/CrBr₃/graphene junction and a change in the low-T CrBr₃ crystal structure when covered by FGT seen in Raman. However, I do not think the authors have convincingly demonstrated proof of a strain-induced moiré as the cause for the experimental observations.

1. The authors claim a “strong coupling between FGT and CrBr₃”. What is the origin of this strong coupling, compared to CrBr₃ combined with graphene or other vdW materials? In other words, what makes FGT special considering that FGT and CrBr₃ have very different lattice constants? It seems unlikely that they would attempt to lattice-match across the vdW gap. How would a difference in thermal expansion coefficients lead the CrBr₃ to adopt a strain gradient to match at the vdW interface upon cooling?
2. The performed measurements are global and could also be explained by other causes. CrBr₃ has been previously shown by the authors to adopt other metastable stackings with AFM order [Ref. 28], seen in graphene/CrBr₃/graphene tunnel junctions and confirmed with Raman. Here, the authors claim that Raman measurements of CrBr₃ under FGT suggest monoclinic stacking, which has AFM order. Can the authors rule out the possibility that FGT induces M stacking in one domain of the CGT flake while retaining R stacking in other parts of the flake? This explanation seems consistent with seeing a coexistence of AFM & FM order in the tunnel junction and a change in the Raman spectrum, without invoking a discussion of moiré superlattices.
3. In light of the above question, what is the spatial resolution of the Raman spectrometer and would it be possible to resolve a map of the shifted peaks to see uniform changes across the FGT-covered CrBr₃ region? This could rule out the possibility of large domains with different stackings/magnetic order.
4. Furthermore, they report that the Raman spectra are not altered at room temperature but are different at low temperature. Is there a transition temperature in which this occurs, and is it associated with a transition in the electrical tunneling data?
5. Could the authors comment on the possibility of a completely alternative explanation based on magnetic interactions occurring at the FGT/CrBr₃ interface at low temperatures, thereby altering the magnetic order in the topmost layer(s) of CrBr₃

but retaining FM order in the remaining layers of the vertical tunnel junction?

6. The discussion of a moiré pattern involves a relevant moiré length scale. The authors do not report any such size. In the picture of moiré magnetism presented, if it were to be the origin of the experimental observations, what is the size of the periodicity between the domains?

Reviewer #2

(Remarks to the Author)

The manuscript describes the coexistence of AFM and FM states in multilayered CrBr₃ induced by differential strain when stacking CrBr₃ with FGT. The authors use low-temperature tunneling current measurements and Raman spectroscopy to confirm this coexistence. They propose a theoretical model explaining the distribution of AFM and FM states in different regions of CrBr₃, attributing this phenomenon to moiré magnetism. While the findings are interesting, questions remain about whether the data provide conclusive evidence for moiré magnetism: The authors suggest that the positive magnetoconductance background (bottom panel of Fig. 2f) resembles that of an AFM CrBr₃ barrier. However, they observe similar positive magnetoconductance of 4% (Fig. 3b) in a pure FM CrBr₃ barrier with both sides contacted by graphene electrodes, which increases to approximately 40% near the Curie temperature. Hence, it is challenging to ascertain from only two devices presented in the paper whether a ~12% positive magnetoconductance (Fig. 3e) arises from AFM state generation in CrBr₃ or from other effects induced by FGT electrodes, which, being an itinerant ferromagnet, can introduce distinct interface magnetic coupling and spin-polarized electrons into the device compared to graphene contact. Even if AFM states are indeed generated in CrBr₃ within the FGT/CrBr₃/Gr device, there is no direct experimental evidence in the transport measurement demonstrating that this is due to moiré magnetism induced by differential strain. (For instance, exchange interactions at the FGT/CrBr₃ interface might govern the magnetization of CrBr₃ layers near the interface and create AFM-like interlayer coupling between specific CrBr₃ layers.) Several groups have already reported systematic modulation of spin configuration in magnetic materials by strain (2020 Adv. Mater. 32 2004533, 2022 Nat. Nanotechnol. 17, 256–261). Compared with them, this work, with device's performance depending on an uncontrollable strain, does not show significance or novelty in this perspective. A potential relevance of this study is to highlight strain-induced moiré magnetism as a crucial innovation. Therefore, additional direct experimental evidence is essential to validate the presence of moiré magnetism. I cannot recommend its publication in Nature Communications as long as this issue is not fully substantiated. Below are other comments and suggestions:

1. The green curves in Fig. 4b exhibit some background signal or noise around the wavenumber range of 130-135 cm⁻¹. Could this be causing the Raman peak originally at ~142 cm⁻¹ to shift leftward? Additionally, the authors only present Raman data at 20 K and room temperature. It would be beneficial to know if they conducted temperature-dependent Raman measurements, particularly at temperatures close to the Curie temperature.
2. If the magnetoconductance is attributed to the AFM state in CrBr₃ induced by differential strain, it should exhibit dependence on the thickness of CrBr₃. Can the authors provide data regarding this thickness dependency?
3. The authors claim that the differential strain arises from the different thermal expansion of FGT and CrBr₃. Did the authors also compare the thermal expansion of graphite and CrBr₃ to demonstrate that there will be no differential strain effect at the CrBr₃/Gr interface? Additionally, is there a difference in the differential strain effect when placing FGT on top or at the bottom of CrBr₃? Is the thickness of FGT critical in this context? Lastly, are there alternative materials that could substitute FGT in fabricating the tunnel junction to observe similar effects?

Reviewer #3

(Remarks to the Author)

General comments: Morpurgo, Z. et al. report evidence for the simultaneous presence of ferromagnetic and antiferromagnetic regions in CrBr₃ by integrate two different electrodes with differential strain, and characterize in detail the correspondence between interfacial strain and magnetic behavior by Raman measurement. The results shown by the authors will be very helpful for the development of this research area and I consider that it is worthy of publication in Nature Communications. And I have several concerns:

Comment 1. Considering that the local magnetic variation depends mainly on the difference in electrode integration, it is necessary for the authors to show the details of the sample preparation process and discuss whether there is annealing behavior after heterojunction stacking, or surface modification by the AFM probe, and the potential impact on the final device behavior.

Comment 2. Through Raman measurement, the authors show that the change in magnetic behavior is mainly due to the difference in CrBr₃ crystal strain (which is also the main evidence of this paper), so the authors test the Raman signal of the material below and next to the FGT in Figure 4 for comparison. However, I think it would be a fairer and more convincing comparison to test the Raman signal of the CrBr₃ material under FGT and under graphene simultaneously.

Comment 3. The authors show that the generation of CrBr₃ strain is mainly due to the difference in thermal conductivity between the FGT electrode and under CrBr₃ material, so it is necessary to characterize the lattice strain and the electrical behavior of the CrBr₃ material with more detailed temperature gradient to support this conclusion, rather than only the two limit temperatures of 2K and room temperature.

Comment 4. For Figure 4, we can see that there is a clear shift at 140 cm⁻¹ peaks, mainly close to the normal Raman peak

resolution (~ 1 cm⁻¹). However, for another peak at 150 cm⁻¹, how did the author accurately distinguish the changes in peak positions at different locations (Fig. 4d)? Perhaps the author gives an explanation or discussion.

Comment 5. For the CrBr₃ strain under FGT electrode, the authors only took a single point test of Raman signal. However, whether the strain is uniform is equally important, so it is necessary to show Raman mapping and give it a discussion.

Reviewer #4

(Remarks to the Author)

Moiré magnetism in CrBr₃ multilayers emerging from differential strain

General comments: Morpurgo, A. et al. report evidence for the simultaneous presence of ferromagnetic and antiferromagnetic regions in CrBr₃ by integrate two different electrodes with differential strain, and characterize in detail the correspondence between interfacial strain and magnetic behavior by Raman measurement. The results shown by the authors will be very helpful for the development of this research area and I consider that it is worthy of publication in Nature Communications. And I have several significant concerns:

Comment 1. Considering that the local magnetic variation depends mainly on the difference in electrode integration, it is necessary for the authors to show the details of the sample preparation process and discuss whether there is annealing behavior after heterojunction stacking, or surface modification by the AFM probe, and the potential impact on the final device behavior.

Comment 2. Through Raman measurement, the authors show that the change in magnetic behavior is mainly due to the difference in CrBr₃ crystal strain (which is also the main evidence of this paper), so the authors test the Raman signal of the material below and next to the FGT in Figure 4 for comparison. However, I think it would be a fairer and more convincing comparison to test the Raman signal of the CrBr₃ material under FGT and under graphene simultaneously.

Comment 3. The authors show that the generation of CrBr₃ strain is mainly due to the difference in thermal conductivity between the FGT electrode and under CrBr₃ material, so it is necessary to characterize the lattice strain and the electrical behavior of the CrBr₃ material with more detailed temperature gradient to support this conclusion, rather than only the two limit temperatures of 2K and room temperature.

Comment 4. For Figure 4, we can see that there is a clear shift at 140 cm⁻¹ peaks, mainly close to the normal Raman peak resolution (~ 1 cm⁻¹). However, for another peak at 150 cm⁻¹, how did the author accurately distinguish the changes in peak positions at different locations (Fig. 4d)? Perhaps the author gives an explanation or discussion.

Comment 5. For the CrBr₃ strain under FGT electrode, the authors only took a single point test of Raman signal. However, whether the strain is uniform is equally important, so it is necessary to show Raman mapping and give it a discussion.

Version 1:

Reviewer comments:

Reviewer #1

(Remarks to the Author)

The authors made significant improvements to the manuscript following the referee comments. They included more thorough Raman data including spatial mapping and temperature dependence, added comparison to a twisted CrBr₃ tunnel junction (that has moire magnetism), and revised the discussion to be more clear in how they interpret their data.

With these changes, I now support publication of this work and believe it will be of great interest to the communities of vdW magnetism and moire materials.

Reviewer #2

(Remarks to the Author)

The authors have created and characterized a twist-aligned CrBr₃ device to compare it with the magnetotransport results from the FGT-CrBr₃ tunnel junction. Additionally, they conducted temperature-dependent Raman measurements and Raman mapping to support their assertion regarding the strain effect. They have addressed the majority of my concerns.

Reviewer #3

(Remarks to the Author)

Authors have made a good effort in replying the previous comments, and I recommend its publication.

Reviewer #4

(Remarks to the Author)

To all Reviewers

We sincerely thank the referees for their time, effort, and valuable comments, which have contributed to significantly improve the quality of our manuscript. In particular, some of the referees wanted to have stronger evidence that the magnetoconductance that we measure in FGT/CrBr₃/Gr barriers originates from moiré magnetism. The referees also raised some questions about some aspects of our Raman measurements, for which they wanted to have more systematic data.

The resubmitted version of the manuscript includes substantial new work that addresses these points in depth. The main additions and changes are:

1. Measurements of the magnetoconductance of three barriers formed by small-angle twisted CrBr₃ ferromagnetic multilayers, where moiré magnetism is expected to occur, to confirm that the observed magnetoconductance in FGT/CrBr₃/Gr barriers does originate from moiré magnetism. We find that the magnetoconductance observed in these twisted barriers reproduces qualitatively all the aspects of the magnetoconductance observed in FGT/CrBr₃/Gr barriers. The similar behavior of the two types of barrier demonstrates that also in the FGT/CrBr₃/Gr devices moiré magnetism occurs. A new Figure has been added to the manuscript text and a related discussion is included in the main text.
2. Raman mapping of CrBr₃ flake on both the FGT-covered and Gr-covered regions, to investigate the spatial extent of differential strain. The related data and discussion confirm that strain is present only under FGT, in agreement with the fact that the magnetoconductance background is also only present in FGT/CrBr₃/Gr barriers. We have added a new supplementary figure with the Raman mapping.
3. Systematic temperature-dependent Raman measurements, to check how differential strain evolves with temperature, and rule out magnetic interaction as the cause of the observed signal. The data substantiate that the strain in CrBr₃ under FGT increases gradually upon lowering temperature, without any sign of structural or magnetic transitions, as expected for the scenario of differential strain proposed in the manuscript. New data are included in Supplementary Fig. 6.

We have also added other data and addressed other specific concerns in our point-by-point replies. We hope that with all the added material, the new version of our manuscript addresses satisfactorily the concerns of the referees and can be accepted for publication.

Reply to Reviewer #1:

Comment 1: *In this work, the authors fabricate 2D tunnel junctions using a FM electrode FGT and magnetic tunnel barrier CrBr₃. Surprisingly, they find evidence for a coexistence of FM and AFM behavior in tunnel junctions using FGT, but not in tunnel junctions using only graphene electrodes (as a control) where CrBr₃ shows the expected FM behavior. This implies that the presence of FGT is somehow influencing the magnetism in CrBr₃, caused by structural effects and/or magnetic interactions.*

They further use Raman spectroscopy of a CrBr₃ flake partially covered by FGT to reveal a structural difference between the bare and FGT-covered regions of CrBr₃ emerging only at low temperature, but not at room temperature. They attribute this change to an induced strain from the difference in thermal expansion coefficients between FGT and CrBr₃, which creates a local moiré superlattice with different symmetries (i.e. locally breaking the C₃ symmetry normally present in AB-stacked rhombohedral CrBr₃). Other stackings in CrBr₃ such as AA and M configurations have AFM coupling, so a strain-induced moiré structure encompassing these varied stackings would lead to the coexistence of AFM and FM order in the tunnel junction.

I agree that the data show a coexistence of FM and AFM order in the FGT/CrBr₃/graphene junction and a change in the low-T CrBr₃ crystal structure when covered by FGT seen in Raman. However, I do not think the authors have convincingly demonstrated proof of a strain-induced moiré as the cause for the experimental observations.

Reply 1: We thank the reviewer for correctly summarizing the main point of the paper, namely the coexistence of FM and AFM behaviour in CrBr₃ multilayers in contact with FGT. The reviewer also has some concerns and questions about the occurrence of a strain-induced moiré structure, which we address point-by-point in the following.

Comment 2: *1. The authors claim a “strong coupling between FGT and CrBr₃”. What is the origin of this strong coupling, compared to CrBr₃ combined with graphene or other vdW materials? In other words, what makes FGT special considering that FGT and CrBr₃ have very different lattice constants? It seems unlikely that they would attempt to lattice-match across the vdW gap. How would a difference in thermal expansion coefficients lead the CrBr₃ to adopt a strain gradient to match at the vdW interface upon cooling?*

Reply 2: We agree with the referee that referring to a “strong coupling between FGT and CrBr₃” is inappropriate. We have modified the discussion in the resubmitted version of the manuscript to avoid misleading the reader in

any way.

We certainly do not claim that the two materials (FGT and CrBr₃) in the barriers that we study try to lattice match one to the other at the interface. What we have in mind is that –as at virtually all interfaces between a multilayer of two different 2D materials– there is a mutual strain induced from one material into the other, irrespective of the precise values of the lattice constants. The strain present at interfaces has a certain strength and relaxes away from the interface (in the direction perpendicular to the layers, causing differential strain) in a way that depends on the specific materials considered, but it appears ubiquitously, and the only questions are of a quantitative nature (i.e., how large the strain is). In CrBr₃, differential strain causes local variation in the interlayer magnetic coupling, which becomes ferro or antiferromagnetic depending on position, as we observe experimentally. The effect is expected to be there for all strengths of differential strain, as the strength only determines the lateral length scale over which the interlayer coupling changes from ferro to antiferromagnetic. For instance, if the differential strain is 0.1% it will take $\sim 1'000$ lattice constants –that is a few hundred nm– to complete a moiré period.

The basic aspects of this scenario are validated by the experiments. In the resubmitted version of the manuscript, we show Raman mapping data on an area where CrBr₃ is covered by either FGT or graphene (Fig.R1 a, b). We observe a distinct Raman shift of the CrBr₃ peaks in the region where CrBr₃ is covered by FGT and no shift in the region where CrBr₃ is covered by graphene. This observation confirms that under the FGT electrode –and not under graphene– the structure of the CrBr₃ multilayer is modified from the pristine rhombohedral (ferromagnetic) stacking of CrBr₃ ferromagnetic multilayers, consistently with the fact that a background magnetoconductance due to moiré is only seen in the region under FGT. We have further carefully checked that the Raman shift of the CrBr₃ peaks in the region under FGT evolves gradually upon cooling: it is very small between room temperature and 100 K, and becomes significant only well below 100 K. We have also checked that the CrBr₃ Raman peaks shift also in “inverted” structures in which CrBr₃ is on top of FGT (shown in Supplementary Fig. 7, not on the bottom as in the devices on which we measured transport). These observations support the scenario that we propose, namely that differential strain emerges progressively as the temperature decreases, driven by the mismatch in thermal expansion coefficients between FGT and CrBr₃.

We have included the spatial mapping and temperature-dependent Raman data in Supplementary Fig. 6 and the Raman data of the inverted structure in Supplementary Fig. 7 of the revised manuscript, along with the related discussion.

Fig.R1. Spatial Raman Mapping of CrBr₃ under FGT and under graphene. **a**, Optical image of a van der Waals heterostructure that contains regions where a CrBr₃ multilayer (outlined by the white dotted line) is covered by a FGT multilayer (outlined by the black dashed line) and by a graphene strip (outlined by a black dotted line). The rectangle delimited by the dashed purple line corresponds to the area over which the Raman signal has been mapped. The position of the E_g mode near 140 cm⁻¹ extracted from these measurements is shown in panel **b**. The mapping indicates a rather uniform peak shift in the FGT-covered region, while no shift is observed in the graphene-covered CrBr₃ region (located inside the rectangle delimited by the orange line). No evidence of multiple domains with varying stacking configurations can be seen in the whole map. Measurements were conducted under parallel (XY) polarization at ~20 K. **c**, Line scan of Raman peak positions extracted from the spatial mapping in panel **b**, taken along the white dashed line, quantifying the peak shift (ΔP) in the FGT-covered CrBr₃ region. **d**, Shift of the position (ΔP) of the E_g mode near 140 cm⁻¹ as a function of temperature (T). The shift is gradual, becomes large below 100 K, and exhibits no sharp features originating from structural or magnetic phase transitions (the dashed line is a guide to the eye).

Comment 3: 2. *The performed measurements are global and could also be explained by other causes. CrBr₃ has been previously shown by the authors to adopt other metastable stackings with AFM order [Ref. 28], seen in graphene/CrBr₃/graphene tunnel junctions and confirmed with Raman. Here, the authors claim that Raman measurements of CrBr₃ under FGT suggest monoclinic stacking, which has AFM order. Can the authors rule out the possibility that FGT induces M stacking in one domain of the CGT flake while retaining R stacking in other parts of the flake? This explanation seems consistent with seeing a coexistence of AFM & FM order in the tunnel junction and a change in the Raman spectrum, without invoking a discussion of moiré superlattices.*

Reply 3: We did consider the possibility of coexisting monoclinic (M) and rhombohedral (AB) stacked domains,

as suggested by the referee, and we can rule that possibility out. The main reason is that M-stacked domains in the CrBr₃ barriers always lead to sharp jumps in the magnetoconductance at 0.55 T or 1.1 T or both. Also the other metastable antiferromagnetic stacking (AA stacking) leads to sharp jumps in the magnetoconductance at 0.2 T or 0.4 T or both (see Fig. R2). In our FGT/ CrBr₃/Gr devices with exfoliated rhombohedral (AB-stacked) CrBr₃, we have never seen equally sharp jumps in the background magnetoconductance (shown in the bottom panel of Fig.2f, g and Fig.3e).

Additionally, to confirm that the magnetoconductance that we observe in FGT/ CrBr₃/Gr barriers originates from moiré magnetism, we have realized and investigated barriers formed by small-angle twisted CrBr₃ ferromagnetic multilayers, in which moiré magnetism is expected to occur. Magnetoconductance measurements on these twisted CrBr₃ barriers show a phenomenology identical to that observed in FGT/ CrBr₃/Gr barriers. We have added Figure.4 and an entire new section to the main text to discuss these new measurements and the comparison between twisted CrBr₃ barriers and FGT/CrBr₃/Gr barriers.

Fig. R2. Low-temperature ($T= 2\text{K}$) tunneling magnetoconductance $\delta G(H)$ of five representative CrBr₃ antiferromagnetic multilayers, showing sharp jumps in at several transition fields (indicated by the vertical grey dashed lines) due to the presence of domains with M- or AA-stacking, for which interlayer coupling is antiferromagnetic.

Comment 4: 3. *In light of the above question, what is the spatial resolution of the Raman spectrometer and would it be possible to resolve a map of the shifted peaks to see uniform changes across the FGT-covered CrBr₃ region?*

This could rule out the possibility of large domains with different stackings/magnetic order.

Reply 4: We thank the referee for this kind suggestion to further rule out the possibility of large antiferromagnetic domains. Following the referee's suggestion, we conducted spatial mapping of Raman spectroscopy (as already shown in Fig. R1), utilizing a laser with a spatial resolution of approximately $\sim 1 \mu\text{m}$ in diameter. We found no evidence for the presence of discrete domains with different stacking.

We now included the Raman spatial mapping data in the Supplementary Fig.6 of the resubmitted manuscript.

Comment 5: *4. Furthermore, they report that the Raman spectra are not altered at room temperature but are different at low temperature. Is there a transition temperature in which this occurs, and is it associated with a transition in the electrical tunneling data?*

Reply 5: Following the referee's suggestion, we performed temperature-dependent Raman measurements from room temperature down to 20 K, taking data every 5 K, and extracted the shift in peak position between the part of CrBr_3 covered by FGT and the part next to the FGT crystal. The result is shown in Fig. R1d. We see that the peak shifts gradually upon cooling, no sharp feature or jump is observed at any temperature (i.e., no indications of a phase transition), and the shift becomes large only well below 100 K. These observations are compatible with the scenario that we propose, namely that the shift is induced by the difference in thermal expansion of CrBr_3 and FGT. Similarly, no transition is observed in the tunneling data upon cooling from room temperature, except for the known magnetic transition of CrBr_3 near 30 K.

We now included the T -dependent Raman data in the Supplementary Fig.6 of the resubmitted manuscript.

Comment 6: *5. Could the authors comment on the possibility of a completely alternative explanation based on magnetic interactions occurring at the FGT/ CrBr_3 interface at low temperatures, thereby altering the magnetic order in the topmost layer(s) of CrBr_3 but retaining FM order in the remaining layers of the vertical tunnel junction?*

Reply 6: A key observation to answer the question of the referee is that the magnetoconductance background that we observe in FGT/ CrBr_3 /Gr barriers is nearly perfectly symmetric as a function of applied bias. This is important because tunneling through these barriers is well described phenomenologically by the Fowler-Nordheim theory, which implies that what contributes the most to transport is the layer near the injecting. Indeed, the spin-valve signal is only seen when injecting electrons from the FGT contact; when injecting electrons from the graphene

contact (and extracting them from the FGT contact), no spin-valve signal is observed. If -as the referee wonders -what we observed was due to magnetic interactions near the FGT contact, the resulting magnetoconductance should only be visible when injecting from FGT, just like it is the case for the spin-valve contribution to the magnetoconductance. The fact that we see a magnetoconductance symmetric upon changing the sign of applied bias (shown in Fig.2f, g and Fig.3e in the main text) means that the phenomenon does not originate from a change occurring near one of the two contacts, but rather somewhere in between the contacts, compatibly with differential strain being responsible for what we observe.

Comment 7: *6. The discussion of a moiré pattern involves a relevant moiré length scale. The authors do not report any such size. In the picture of moiré magnetism presented, if it were to be the origin of the experimental observations, what is the size of the periodicity between the domains?*

Reply 7: The referee is indeed correct: the magnetoconductance data do not provide any direct information about the periodicity of the moiré. We now make this point explicitly clear in the main text.

We expect the periodicity to be large. For instance, a strain in CrBr₃ under FGT of 1% –a realistic, fairly large value– would imply that the differential strain is only a fraction of 1%. If we assume a differential strain of 0.1 %, we would conclude that we would need approximately 1000 unit cells to make a moiré period. This would imply a moiré periodicity of a few hundred nm, much larger than moiré periods in twisted structures. Note, however, that such a differential strain –or even a smaller value– would always cause the coexistence of regions with ferro and antiferromagnetic interlayer coupling over the lateral extension of the junctions (which is several microns by several microns in dimensions), which would explain our observations (both magnetoconductance and Raman).

Reply to Reviewer #2:

Comment 1: *The manuscript describes the coexistence of AFM and FM states in multilayered CrBr₃ induced by differential strain when stacking CrBr₃ with FGT. The authors use low-temperature tunneling current measurements and Raman spectroscopy to confirm this coexistence. They propose a theoretical model explaining the distribution of AFM and FM states in different regions of CrBr₃, attributing this phenomenon to moiré magnetism. While the findings are interesting, questions remain about whether the data provide conclusive evidence for moiré magnetism*

Reply 1: We are pleased to read that the reviewer finds our results to be interesting. Nevertheless, the reviewer also has questions about evidence for moiré magnetism, which we address point-by-point in the following.

Comment 2: *The authors suggest that the positive magnetoconductance background (bottom panel of Fig. 2f) resembles that of an AFM CrBr₃ barrier. However, they observe a similar positive magnetoconductance of 4% (Fig. 3b) in a pure FM CrBr₃ barrier with both sides contacted by graphene electrodes, which increases to approximately 40% near the Curie temperature. Hence, it is challenging to ascertain from only two devices presented in the paper whether a ~12% positive magnetoconductance (Fig. 3e) arises from AFM state generation in CrBr₃ or from other effects induced by FGT electrodes, which, being an itinerant ferromagnet, can introduce distinct interface magnetic coupling and spin-polarized electrons into the device compared to graphene contact.*

Reply 2: We have fabricated a total of four FGT/ CrBr₃/Gr devices and –for this and previous work– many Gr/ CrBr₃/Gr devices. We can therefore analyse the magnitude of the magnetoconductance in both types of devices giving some statistics. To this end, we have summarized the values of magnetoconductance ($T = 2\text{K}$, $H = 1\text{ T}$, average of the two values measured at opposite biases) for FGT/ CrBr₃/Gr devices and compared it to the magnetoconductance of four Gr/CrBr₃/Gr devices. The data show that there are sample-to-sample differences, but all FGT/ CrBr₃/Gr devices exhibit a significantly larger magnetoconductance than all Gr/CrBr₃/Gr devices. If we take the average magnetoconductance over the four devices of each type, we find that at $H = 1\text{ T}$ the magnetoconductance of FGT/CrBr₃/Gr devices is approximately 22%, and that of Gr/ CrBr₃/Gr device approximately 2.1%. FGT/ CrBr₃/Gr therefore exhibit one order of magnitude larger magnetoconductance of Gr/ CrBr₃/Gr devices.

We now include this figure in the Supplementary Fig.4 of the resubmitted manuscript.

Fig. R3. Summary of magnetoconductance values, δG ($H = 1$ T, 2 K), measured in four FGT/CrBr₃/Gr devices and Gr/CrBr₃/Gr devices fabricated using rhombohedral CrBr₃. The average magnetoconductance ($\sim 22\%$) with FGT electrodes is approximately ten times larger than that of devices using only graphene electrodes ($\sim 2.1\%$). In all cases, the CrBr₃ multilayer employed to realize the devices is AB-stacked (i.e., ferromagnetic in its pristine form).

Comment 3: *Even if AFM states are indeed generated in CrBr₃ within the FGT/CrBr₃/Gr device, there is no direct experimental evidence in the transport measurement demonstrating that this is due to moiré magnetism induced by differential strain. (For instance, exchange interactions at the FGT/CrBr₃ interface might govern the magnetization of CrBr₃ layers near the interface and create AFM-like interlayer coupling between specific CrBr₃ layers.)*

Reply 3: We can make two important considerations to address the concern of the referee. The first is to exclude what causes the magnetoconductance that we observe in FGT/CrBr₃/Gr devices originates from magnetic interactions at the FGT/CrBr₃ interface. To this end, we note that the magnetoconductance background that we observe in FGT/CrBr₃/Gr barriers is nearly perfectly symmetric upon changing the polarity of the applied bias (shown in Fig.2f, g and Fig.3e in the main text). This is important because tunneling through these barriers is well described phenomenologically by the Fowler-Nordheim theory, which implies that what contributes the most to transport is the layer near the injecting contact. Indeed, the spin-valve signal is only seen when injecting electrons from the FGT contact; when injecting electrons from the graphene contact (and extracting them from the FGT contact), no spin-valve signal is observed. If -as the referee wonders- what we observed was due to magnetic interactions near the FGT contact, the resulting magnetoconductance should only be visible when injecting from FGT, just like it is the case for the spin-valve contribution to the magnetoconductance. The fact that we see a magnetoconductance symmetric upon changing the sign of applied bias means that the phenomenon does not

originate from a change occurring near one of the two contacts, but rather somewhere in between the contacts, compatibly with differential strain being responsible for what we observe.

The second –more important – consideration is based on new measurements that we have performed to confirm that the background magnetoconductance of FGT/CrBr₃/Gr barriers originates from moiré magnetism. To this end we have realized three tunnel barriers based on small-angle (<3°) twisted CrBr₃ multilayers, creating intentionally an interface in the center of the CrBr₃ barrier where moiré magnetism is indeed expected to occur. The magnetoconductance data of these barriers (see Fig. 4 of the main text in the resubmitted version) show that the behaviour of twisted barriers and FGT/CrBr₃/G barriers is virtually identical. They both show a very similar shape of the magnetoconductance over the same magnetic field range, the absence of sharp jumps (which are present in CrBr₃ antiferromagnetic barriers), symmetry upon reversing the sign of the applied bias, and the same temperature dependence (shown in Supplementary Fig. 3). As FGT/CrBr₃/Gr barriers exhibit magnetoconductance that is virtually identical to that of twisted Gr/CrBr₃/Gr barriers due to moiré magnetism, we conclude that the magnetoconductance of FGT/CrBr₃/Gr is also due to moiré magnetism.

We have added Fig.4 and Supplementary Fig. 3 to the revised manuscript with new data on the magnetoconductance of twisted small-angle CrBr₃ barriers and an entirely new section to discuss this data in the main text. We hope that these data, and the considerations in our reply, address the referee's concerns about the conclusion that the magnetoconductance of FGT/CrBr₃/Gr barriers originates from moiré magnetism.

Comment 4: *Several groups have already reported systematic modulation of spin configuration in magnetic materials by strain (2020 Adv. Mater. 32 2004533, 2022 Nat. Nanotechnol. 17, 256–261). Compared with them, this work, with device's performance depending on an uncontrollable strain, does not show significance or novelty in this perspective. A potential relevance of this study is to highlight strain-induced moiré magnetism as a crucial innovation. Therefore, additional direct experimental evidence is essential to validate the presence of moiré magnetism. I cannot recommend its publication in Nature Communications as long as this issue is not fully substantiated.*

Reply 4: We have followed the recommendation of the referee and included new experimental data (as mentioned above) to confirm the presence of moiré magnetism in FGT/ CrBr₃ /Gr barriers, by comparing the magnetoconductance of these barriers to the magnetoconductance of small-angle (less than 3°) twisted CrBr₃ barriers, in which moiré magnetism is expected. The phenomenology of the two magnetoconductance

measurements in the two different types of barriers is virtually identical, which provides a new, independent, and strong indication that moiré magnetism is at the origin of the magnetotransport response of FGT/CrBr₃/Gr barriers. In addition, in the resubmitted version of the manuscript, we show Raman mapping data on an area where CrBr₃ is covered by either FGT or graphene (shown in Supplementary Fig.6). We observe a distinct Raman shift of the CrBr₃ peaks in the region where CrBr₃ is covered by FGT and no shift in the region where CrBr₃ is covered by graphene. This observation confirms that under the FGT electrode –and not under graphene–the structure of the CrBr₃ multilayer is modified from the pristine rhombohedral (ferromagnetic) stacking of CrBr₃ ferromagnetic multilayers, consistently with the fact that a background magnetoconductance due to moiré is only seen in the region under FGT. We have further carefully checked that the Raman shift of the CrBr₃ peaks in the region under FGT evolves gradually upon cooling: it is very small between room temperature and 100 K, and becomes significant only well below 100 K. We have also checked that the CrBr₃ Raman peaks shift also in “inverted” structures in which CrBr₃ is on top of FGT (shown in Supplementary Fig. 7, not on the bottom as in the devices on which we measured transport). These observations support the scenario that we propose, namely that differential strain emerges progressively as the temperature decreases, driven by the mismatch in thermal expansion coefficients between FGT and CrBr₃.

We have added a new Fig.4 and an entirely new section to the main text to discuss these new measurements and the comparison between twisted CrBr₃ barriers and FGT/CrBr₃/Gr barriers. We also have included the spatial mapping and temperature-dependent Raman data in Supplementary Fig. 6 and the Raman data of the inverted structure in Supplementary Fig. 7 of the revised manuscript, along with the related discussion. The referee mentioned two important references, one was already cited, now we have included both in the revised manuscript.

Comment 5: *Below are other comments and suggestions:*

1. The green curves in Fig. 4b exhibit some background signal or noise around the wavenumber range of 130-135 cm⁻¹. Could this be causing the Raman peak originally at ~142 cm⁻¹ to shift leftward?

Reply 5: The feature in the 130-135 cm⁻¹ range that the referee is commenting about originates from FGT itself. To clarify this point, we compare here over a broader interval of wavelengths the Raman spectrum measured on the FGT/CrBr₃ heterostructure (orange curve in the bottom panel of Fig. R4) with the spectrum measured on FGT alone (purple line in the top panel of Fig. R4). This comparison shows that the peak at 120 cm⁻¹ as well as the shoulder around 130 cm⁻¹ originate from the FGT crystal.

We apologize for any confusion caused by our initial plot. To avoid potential misinterpretation by the other readers, we have reduced the Raman spectra range shown in the current Fig.5 (previously Fig. 4)

Fig. R4. Raman spectra measured at $T = 300$ K on a FGT crystal and on the FGT/CrBr₃ heterostructure (top and bottom panels, respectively), over a broader wavelength interval than the one shown in the manuscript. A peak at 120 cm⁻¹ is present due to FGT itself.

Comment 6: *Additionally, the authors only present Raman data at 20 K and room temperature. It would be beneficial to know if they conducted temperature-dependent Raman measurements, particularly at temperatures close to the Curie temperature.*

Reply 6: We followed the referee's recommendation and performed more systematic Raman measurements as a function of temperature, between 20 K and room temperature, with a 5 K incremental step. From these measurements, we extracted the shift in the position of the CrBr₃ Raman peaks in the FGT-covered region as compared to the CrBr₃ next to the FGT crystal. No sharp change near the Curie temperature is seen in the data.

We now included the T -dependent Raman data in the Supplementary Fig. 6 of the resubmitted manuscript.

Comment 7: *2.If the magnetoconductance is attributed to the AFM state in CrBr₃ induced by differential strain, it should exhibit dependence on the thickness of CrBr₃. Can the authors provide data regarding this thickness dependency?*

Reply 7: We fabricated a total of four devices with FGT, and their normalized magnetoconductance is shown in Fig.4d. In the thickness range we explored (3.4 nm to 8.5 nm), the magnetoconductance exhibits a similar trend: a smooth magnetoconductance response. This suggests that, within this thickness range, the differential strain in the different devices is comparable.

We have added the detailed thickness information of CrBr₃ in the legend of the new Fig.4d.

Comment 8: *3.The authors claim that the differential strain arises from the different thermal expansion of FGT and CrBr₃. Did the authors also compare the thermal expansion of graphite and CrBr₃ to demonstrate that there will be no differential strain effect at the CrBr₃/Gr interface?*

Reply 8: Following the referee's question, to investigate whether there is differential strain at the CrBr₃/Gr interface, we conducted Raman mapping on a region that includes both FGT and graphene (Supplementary Fig. 6 a,b). In the FGT region, a distinct Raman shift and broadening were observed, indicating structural modification of the CrBr₃ multilayers. In contrast, no such shift was detected under graphene contacts, suggesting that graphene does not induce any detectable structural changes in CrBr₃ multilayers.

We now included the related discussion in the resubmitted manuscript and Raman spatial mapping data in Supplementary Fig.6.

Comment 9: *Additionally, is there a difference in the differential strain effect when placing FGT on top or at the bottom of CrBr₃?*

Reply 9: In response to the referee's question, we fabricated a heterostructure with FGT placed at the bottom of CrBr₃ and performed polarized Raman spectroscopy on both CrBr₃/FGT and CrBr₃ (data presented in Supplementary Fig.7). The Raman data reveal a similar trend to when FGT is placed on top of CrBr₃, as shown in Fig. 5 of the main text. Specifically, at low temperatures, we observe peak shifts and broadening in the CrBr₃/FGT heterostructure, indicating that the differential strain effect is comparable regardless of FGT's position.

We now included the Raman data and optical pictures of the inverted heterostructure in the Supplementary Fig.7 of the resubmitted manuscript.

Comment 10: *Is the thickness of FGT critical in this context?*

Reply 10: A thinner FGT is generally preferred, as constructing a heterostructure becomes more challenging with a thicker FGT layer. On the other hand, it is difficult to exfoliate very thin FGT layers. That is why in our work we focus on the range between approximately 5 nm and 10 nm. In this FGT consistently behaves as a ferromagnetic metal, regardless of the exact thickness.

Comment 11: *Lastly, are there alternative materials that could substitute FGT in fabricating the tunnel junction to observe similar effects?*

Reply 11: We appreciate the referee's question and agree that demonstrating similar results with alternative materials would be valuable. In our work, FGT(Fe_3GeTe_2) plays a dual role: (1) it induces differential strain in CrBr_3 due to its distinct thermal expansion coefficient, leading to the emergence of antiferromagnetism in CrBr_3 , and (2) as a ferromagnetic metal, it serves as a probe by demonstrating the spin-valve effect (an indicator of ferromagnetism). While materials like Fe_3GaTe_2 could potentially substitute FGT, a thorough exploration of this possibility would require significant additional work and could be a focus for future studies.

Reply to Reviewer #3:

General comments: *Morpurgo, Z. et al. report evidence for the simultaneous presence of ferromagnetic and antiferromagnetic regions in CrBr₃ by integrating two different electrodes with differential strain, and characterize in detail the correspondence between interfacial strain and magnetic behavior by Raman measurement. The results shown by the authors will be very helpful for the development of this research area and I consider that it is worthy of publication in Nature Communications. And I have several concerns:*

Reply: We thank the referee for the positive comments on our work and for their recommendation to publish our manuscript. The reviewer also has some concerns and questions, which we address point-by-point in the following.

Comment 1. *Considering that the local magnetic variation depends mainly on the difference in electrode integration, it is necessary for the authors to show the details of the sample preparation process and discuss whether there is annealing behavior after heterojunction stacking, or surface modification by the AFM probe, and the potential impact on the final device behavior.*

Reply 1: We would like to clarify that neither an annealing process nor AFM surface modification was performed after heterojunction stacking.

Following the referee's suggestion, we have included a more detailed description of the sample preparation process in the methods section of the revised manuscript. In the assembly process, a PDMS-PC stamp was used to pick up the top h-BN at 90°C, followed by the top graphene, FGT, CrBr₃ and bottom graphene, each at 70°C, and the bottom h-BN at 90°C. The PC with the whole stack was finally released onto a SiO₂/Si substrate at 160°C. After transfer, the substrate was immersed in chloroform to dissolve the PC, leaving the heterostructure on the substrate.

Comment 2. *Through Raman measurement, the authors show that the change in magnetic behavior is mainly due to the difference in CrBr₃ crystal strain (which is also the main evidence of this paper), so the authors test the Raman signal of the material below and next to the FGT in Figure 4 for comparison. However, I think it would be a fairer and more convincing comparison to test the Raman signal of the CrBr₃ material under FGT and under graphene simultaneously.*

Reply 2: In the resubmitted version of the manuscript, we show Raman mapping data on an area where CrBr₃ is covered by either FGT or graphene (Supplementary Fig.6). We observe a distinct Raman shift of the CrBr₃ peaks in the region where CrBr₃ is covered by FGT, and no shift in the region where CrBr₃ is covered by graphene. This observation confirms that under the FGT electrode –and not under graphene– the structure of the CrBr₃ multilayer is modified from the pristine rhombohedral (ferromagnetic) stacking of CrBr₃ ferromagnetic multilayers, consistently with the fact that a background magnetoconductance due to moiré is only seen in the region under FGT.

Comment 3. *The authors show that the generation of CrBr₃ strain is mainly due to the difference in thermal conductivity between the FGT electrode and under CrBr₃ material, so it is necessary to characterize the lattice strain and the electrical behavior of the CrBr₃ material with more detailed temperature gradient to support this conclusion, rather than only the two limit temperatures of 2K and room temperature.*

Reply 3: We have further carefully checked how the shift of the Raman peaks of CrBr₃ evolves as a function of temperature (as shown in the Supplementary Fig.6). The relative Raman shift between the FGT-covered and uncovered parts of CrBr₃ evolves gradually upon cooling. The shift is vanishingly small at room temperature and increase –but remains small– down to 100 K, and it becomes significant as the temperature is lowered well below 100 K. The evolution is smooth, without any sharp features or jump. These observations support the scenario that we propose, namely that differential strain emerges progressively as the temperature decreases, driven by the mismatch in thermal expansion coefficients between FGT and CrBr₃.

Comment 4. *For Figure 4, we can see that there is a clear shift at 140 cm⁻¹ peaks, mainly close to the normal Raman peak resolution (~1 cm⁻¹). However, for another peak at 150 cm⁻¹, how did the author accurately distinguish the changes in peak positions at different locations (Fig. 4d)? Perhaps the author gives an explanation or discussion.*

Reply 4: Following the referee's suggestion, we have added a more detailed discussion in the methods section of the revised manuscript to clarify this point. Specifically, all measured Raman spectra were fitted with a set of Voigt functions (Gaussian-Lorentzian convolution, detail in reference: *J. Raman Spectrosc.* 20,359-365 (1989).) to accurately resolve the peak positions, even for closely spaced peaks. By optimizing the fitting parameters, we minimized errors arising from the limited resolution (~ 0.1 cm⁻¹), ensuring precise extraction of peak positions.

Comment 5. For the CrBr₃ strain under FGT electrode, the authors only took a single point test of Raman signal. However, whether the strain is uniform is equally important, so it is necessary to show Raman mapping and give it a discussion.

Reply 5: Following the referee's suggestion, we performed spatial Raman mapping using a laser with a spatial resolution of approximately 1 μm. As the laser moves gradually from the uncovered region to the FGT-covered area, a clear peak shift is observed. The Raman data indicates uniform changes across the mapped region. This allows us to exclude that domains with different stacking (e.g., the known metastable antiferromagnetic phases of CrBr₃) are present under the FGT contact.

We now included the Raman spatial mapping data in the Supplementary Fig.6 of the resubmitted manuscript.

Reply to Reviewer #1:

Comment: The authors made significant improvements to the manuscript following the referee comments. They included more thorough Raman data including spatial mapping and temperature dependence, added comparison to a twisted CrBr₃ tunnel junction (that has moire magnetism), and revised the discussion to be more clear in how they interpret their data.

With these changes, I now support publication of this work and believe it will be of great interest to the communities of vdW magnetism and moire materials.

Reply: We sincerely thank the referee for recommending our manuscript for publication. We are grateful for the time, effort, and valuable comments, which have contributed to significantly improve the quality of our work.

Reply to Reviewer #2:

Comment: The authors have created and characterized a twist-aligned CrBr₃ device to compare it with the magnetotransport results from the FGT-CrBr₃ tunnel junction. Additionally, they conducted temperature-dependent Raman measurements and Raman mapping to support their assertion regarding the strain effect. They have addressed the majority of my concerns.

Reply: We sincerely thank the referees for their time, effort, and valuable comments, which have contributed to significantly improve the quality of our manuscript.

Reply to Reviewer #3:

Comment: Authors have made a good effort in replying the previous comments, and I recommend its publication.

Reply: We sincerely thank the referee for recommending our manuscript for publication. We are grateful for the time, effort, and valuable comments, which have contributed to significantly improve the quality of our work.

Reply to Reviewer #4:

Comment: I co-reviewed this manuscript with one of the reviewers who provided the listed reports. This is part of the Nature Communications initiative to facilitate training in peer review and to provide appropriate recognition for Early Career Researchers who co-review manuscripts.

Reply: We sincerely thank the referees for their time, effort, and valuable comments, which have contributed to significantly improve the quality of our manuscript.